# Genetics of circulating inflammatory proteins identifies drivers of immune-mediated disease risk and therapeutic targets

Circulating proteins have important functions in inflammation and a broad range of diseases. To identify genetic influences on inflammation-related proteins, we conducted a genome-wide protein quantitative trait locus (pQTL) study of 91 plasma proteins measured using the Olink Target platform in 14,824 participants. We identified 180 pQTLs (59 *cis*, 121 *trans*). Integration of pQTL data with eQTL and disease genome-wide association studies provided insight into pathogenesis, implicating lymphotoxin-α in multiple sclerosis. Using Mendelian randomization (MR) to assess causality in disease etiology, we identified both shared and distinct effects of specific proteins across immune-mediated diseases, including directionally discordant effects of CD40 on risk of rheumatoid arthritis versus multiple sclerosis and inflammatory bowel disease. MR implicated CXCL5 in the etiology of ulcerative colitis (UC) and we show elevated gut *CXCL5* transcript expression in patients with UC. These results identify targets of existing drugs and provide a powerful resource to facilitate future drug target prioritization.

Inflammation is a physiological host response to infection or injury. However, aberrant inflammatory responses result in tissue damage and are central to the pathogenesis of multiple diseases, including sepsis, autoimmunity and atherothrombosis. Inflammatory responses are orchestrated by a complex network of cells and mediators, including circulating proteins such as cytokines and soluble receptors. Therefore, discovery of the genetic determinants of the abundance of inflammation-related circulating proteins should yield valuable insights into both physiology and the etiology of a broad range of diseases.

Proteomic studies are informative because proteins are the effector molecules of most biological processes and, from a translational perspective, proteins are the targets of most drugs. The development of high-throughput proteomic technologies now allows for profiling of the plasma proteome on an epidemiological scale. Coupling genomic and proteomic data enables identification of genetic variants associated with protein abundance, pQTLs. pQTLs provide valuable insights into the molecular basis of complex traits and diseases by identifying proteins that lie between genotype and phenotype. Recent years have seen a rapid increase in both the number and the size of pQTL studies, transforming our understanding of the genetic architecture of the circulating proteome[1–11].

In the present study, we extend previous work by performing pQTL mapping for 91 inflammation-related proteins in 14,824 participants. We integrate these data with disease genome-wide association studies (GWASs) to characterize the functional effects of disease-associated variants. Using MR and colocalization analyses, we identified proteins that play a causal role in immune-mediated disease etiology. Our results revealed pathways that are known to be therapeutically important and new putative drug targets, including CD40 in rheumatoid arthritis, lymphotoxin-α (LTA) in multiple sclerosis and the chemokine CXCL5 in UC.

✉e-mail: asb38@medschl.cam.ac.uk; j.peters@imperial.ac.uk

## Results

### Genetic architecture of circulating inflammatory proteins

We performed genome-wide pQTL mapping for 91 plasma proteins measured using the Olink Target Inflammation panel in 11 cohorts totaling 14,824 European-ancestry participants (Supplementary Table 1 and Supplementary Note 1) and meta-analyzed the results (Extended Data Fig. 1). To provide a succinct and standardized nomenclature, we report proteins using the non-italicized symbols of the genes encoding them (see Supplementary Table 2 for a mapping of symbols to full protein names and UniProt identifiers). We identified a total of 180 significant ($P \leq 5 \times 10^{-10}$, fixed-effect meta-analysis) associations between 108 genomic regions (see Methods for locus definition) and 70 proteins (Fig. 1, Supplementary Table 3, Supplementary Item and Supplementary Figs. 1 and 2). Of the 180 significant locus–protein associations, 59 (33%) were local acting ('*cis*' pQTLs; defined here as a genetic variant lying within ±1 Mb of the gene encoding the associated protein) and 121 (67%) were distant acting ('*trans*'). We found evidence of *trans*-pQTL hotspots associated with multiple proteins (for example, rs3184504 at the *SH2B3* locus was associated with CXCL9, CXCL10, CXCL11, CD5, CD244 and IL-12B) (Fig. 2a).

For 70 (77%) of the 91 proteins studied, we identified at least 1 significant pQTL, including 59 (65%) proteins that had a *cis*-pQTL. Of these 70 proteins, 19 had only a *cis*-pQTL, 11 had only *trans*-pQTL(s) and 40 had both *cis*- and *trans*-pQTLs. For 18 of the 21 proteins for which no pQTL was detected, >50% of samples had levels below the lower limit of detection (LLOD), suggesting that the lack of genetic signal is due to low protein abundance in plasma (Extended Data Fig. 2a). The number of genomic loci associated with each protein ranged between one and eight (Fig. 2b), but was fewer than four for most proteins. Examples of multi-locus-regulated proteins include IL-12B and TNFSF10, both of which had one *cis*- and seven *trans*-pQTLs (Fig. 2c,d). Conditional analyses revealed the presence of an additional 47 independent signals, raising the total number of pQTL signals from 180 (59 *cis*, 121 *trans*) to 227 (99 *cis*, 128 *trans*) (Supplementary Table 4).

To validate our pQTL results, we tested significant associations from our discovery meta-analysis for replication in an independent cohort (ARISTOTLE) comprising 1,585 participants with Olink plasma proteomic data[12]. Of the 180 pQTL signals, we were able to test 174 in the ARISTOTLE data, of which 168 had a directionally consistent effect estimate. There was a strong correlation (Pearson's $r = 0.97$) between the pQTL effect estimates in ARISTOTLE and those in the discovery meta-analysis; this correlation was consistent for both *cis*- and *trans*-pQTL effect sizes ($r = 0.99$ and $r = 0.94$, respectively) (Extended Data Fig. 3). Out of the 174 pQTL signals, 32 replicated at $P \leq 5 \times 10^{-10}$ (linear regression) and 72 at $P \leq 2.8 \times 10^{-4}$ (a Bonferroni-corrected threshold), respectively (Supplementary Table 5). We also tested our significant pQTLs for replication in 35,556 Icelanders from the deCODE study[9], which assayed plasma proteins using the aptamer-based SomaScan platform (Supplementary Note 2). Of the 91 proteins in our study, 72 were measured in the deCODE study. Of the 158 locus–protein associations that could be tested, 75 were significant at $P \leq 5 \times 10^{-10}$ (linear regression) and 96 at $P \leq 2.8 \times 10^{-4}$. Overall, we replicated 126 (71%) of the 178 testable pQTLs in either ARISTOTLE or deCODE at $P \leq 2.8 \times 10^{-4}$ (linear regression) (Supplementary Note Table 1).

In line with other GWASs, we observed an inverse relationship between effect size and minor allele frequency (MAF), with rarer pQTL variants generally showing larger effect sizes (Extended Data Fig. 4a). The proportion of variance explained by the significant sentinel variants from our discovery meta-analysis varied from 0.003 for NTF3 to 0.285 for CCL8 (Extended Data Fig. 4b).

### Annotation and characterization of *cis*-pQTLs

Of the 59 *cis*-pQTLs identified, 11 sentinel variants were protein-altering variants (PAVs) (10 missense and 1 splice acceptor). A further 10 sentinel variants were in high linkage disequilibrium (LD; $r^2 > 0.8$) with a protein-altering variant (all missense). Of these, seven were variants in the gene encoding the target protein itself and three in another nearby gene (Supplementary Note 3). PAVs can result in false-positive *cis*-pQTL signals by altering protein epitopes recognized by antibodies used in proteomic assays[13]. However, they can also impact the abundance of plasma proteins through several mechanisms, including protein translation, secretion into the circulation, enzymatic cleavage of pre-proteins and protein clearance and degradation. Alternatively, plasma protein abundance can also be affected by altered transcriptional regulation in blood cells or other tissues.

We next examined the degree to which the 59 *cis*-pQTLs were explained by corresponding *cis*-expression (e)QTLs, by comparing our findings with publicly available *cis*-eQTL data. In a meta-analysis of whole-blood eQTL data from the eQTLGen Consortium[14], we found a genome-wide significant ($P \leq 5 \times 10^{-8}$; meta-analysis) *cis*-eQTL for 40 of the 59 *cis*-pQTLs, where the *cis*-eQTL target gene encodes the *cis*-pQTL target protein. However, statistical colocalization analyses showed that only 6 (rs34790908-*TNFSF12*, rs72912115-*TGFA*, rs471994-*MMP1*, rs674379-*CD5*, rs450373-*CXCL5* and rs5744249-*IL18*) of these *cis*-eQTLs colocalized (posterior probability (PP) $\geq 0.8$) with their cognate *cis*-pQTLs (Supplementary Table 6), indicating that the remaining 34 eQTL–pQTL pairs may not share the same underlying causal genetic variant. Examination of regional association plots confirmed that most blood eQTL and pQTL signals were distinct (Supplementary Fig. 3). Of the six colocalizing eQTL–pQTL pairs, five were directionally consistent. However, the eQTL and pQTL for IL18 at rs5744249 were oppositely associated with the messenger RNA and protein levels. rs5744249 resides in intron 2 of *IL18* and is in high LD with a 3′-UTR variant (rs5744292, $r^2 = 0.98$, 1000 Genomes EUR), but no PAVs. Therefore, the directional discordance is not easily explained either by an artefactual pQTL signal due to altered antibody binding or by a difference in the release of IL-18 into the circulation due to differences in protein structure, but may instead relate to differential post-transcriptional regulatory mechanisms or contributions of different cell types to the plasma pQTL versus whole-blood eQTL. Indeed, directional uncoupling of eQTL–pQTL pairs has been previously reported[8] and eQTL directional discordance has been observed between different tissues[15] or even within different leukocytes[16].

As tissues other than blood are the primary source of many plasma proteins, we explored eQTL data across a range of tissues and cell types from the Genotype Tissue Expression (GTEx) (v.8) project[15] and the eQTL Catalogue[17]. Systematic colocalization analyses revealed colocalizing (PP $\geq 0.8$) *cis*-eQTLs in at least one tissue or cell type for 32 of the 59 *cis*-pQTLs (Supplementary Tables 7 and 8): 16 were highlighted by both eQTL resources, 12 by GTEx only and the remaining 4 by the eQTL Catalogue only. These included all six colocalizing *cis*-eQTLs from eQTLGen. These findings suggest that at least 50% of the *cis*-pQTLs identified in our study may be driven by underlying cognate *cis*-eQTLs. In most cases, colocalization (PP $\geq 0.8$) between *cis*-eQTL and pQTL pairs was observed across two or more distinct tissues or cell types, up to a maximum of 41 (for rs1883832-*CD40*). In other cases, colocalization was observed in just a single tissue or cell type (for example, the colocalizing *cis*-eQTL signal at rs62360376 for *GDNF* was found only in skeletal muscle). Of the 27 *cis*-pQTLs without a corresponding colocalizing *cis*-eQTL, for 12 the sentinel variant or a proxy in high LD was a PAV (see Supplementary Note Table 3).

### Identifying the mediators of *trans*-pQTLs

We sought to identify the most likely gene mediators of the *trans*-pQTLs using the ProGeM bioinformatics tool[18], which utilizes genomic (for example, *cis*-eQTL) and biological (for example, gene ontology (GO) and pathways) annotation data from multiple sources. For some *trans*-pQTLs, we identified strong evidence to implicate a gene encoded near the pQTL as mediating the distant association with the target protein. Examples included receptor–ligand pairs such as IL-6–IL-6R, IL-10–IL-

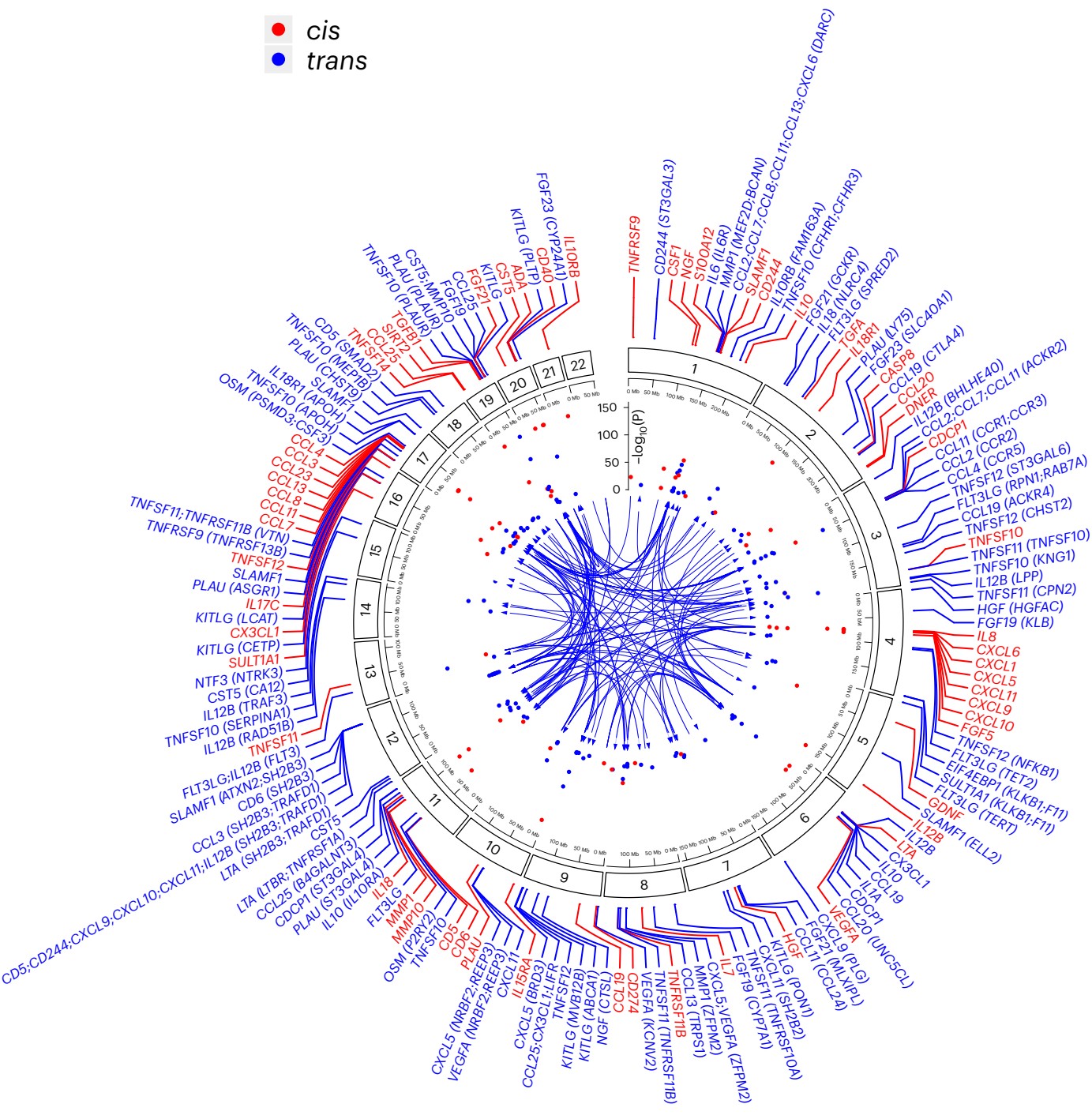

**Fig. 1 | Genomic map of genetic determinants of inflammation-related proteins.** Circos plot linking the location of pQTLs to the gene encoding their associated proteins. Labels for the *cis*-pQTLs (red) indicate the gene encoding the target protein. For the *trans*-pQTLs (blue), the gene symbols of the target proteins are indicated, along with the putative mediating gene(s) at the *trans*-pQTLs in brackets where applicable. The $-\log_{10}(P)$ values are capped at 150 for readability. Two-sided *P* values are from meta-analysis of linear regression estimates.

10RA, CCL2–CCR2, CCL4–CCR5 and CCL11–CCR3. We also identified genes mediating pQTLs through intracellular signaling pathways rather than direct ligand–receptor interactions. An example is rs385076, an intronic variant in *NLRC4*, which is a *trans*-pQTL for IL-18. IL-18 is synthesized as an inactive precursor (pro-IL-18), which is cleaved by caspase-1 in the NLRC4 inflammasome to produce the active form of IL-18 (Fig. 3a). As rs385076 is also a *cis*-eQTL for the inflammasome gene *NLRC4* (Fig. 3b), together these QTL data suggest that genetic variation

in *NLRC4* alters its expression and thereby inflammasome activity, with consequent effects on circulating IL-18 levels.

Following a manual literature review to refine the ProGeM output, we narrowed down the most likely mediating gene(s) to either 1 or 2 candidates for 100 of the 121 *trans*-pQTLs (Supplementary Table 9). For 94, 1 of the 3 nearest genes to the sentinel variant was the primary candidate. In several instances where either one or two candidate genes were prioritized, ProGeM revealed functional links between both (1)

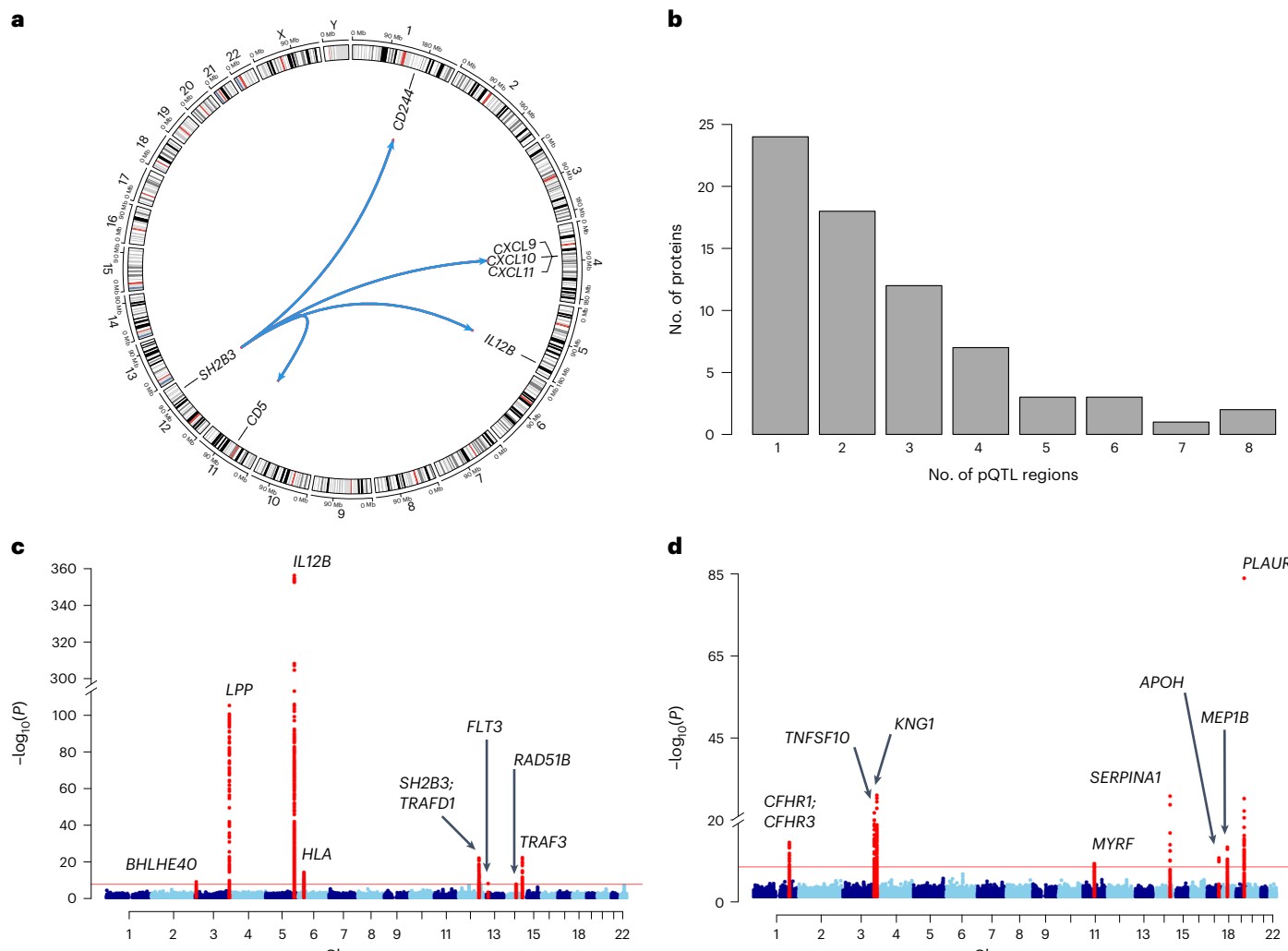

**Fig. 2 | Genetic architecture of 91 inflammation-related proteins. a**, Circos plot showing the *trans*-pQTL 'hotspot' at the *SH2B3* locus on chromosome 12, associated with six proteins. **b,** Distribution of the number of identified pQTLs per protein. The *HLA* was treated as a single region. **c,d**, Manhattan plots showing genetic associations with plasma abundance of IL-12B (**c**) and TNFSF10 (TRAIL) (**d**). The horizontal red line indicates statistical significance ($P = 5 \times 10^{-10}$). Two-sided $P$ values are from meta-analysis of linear regression estimates. The nearest genes in the region of pQTL signals are annotated.

the sentinel variant and the nearby candidate mediating gene (for example, *cis*-eQTL) and (2) the same candidate mediating gene and the *trans*-affected protein(s) (for example, through protein–protein interaction). We have previously shown that such convergence on the same gene is indicative of a strong candidate[18]. An example of this is the *trans*-pQTL at rs12075, which is associated with multiple chemokines (CCL2, CCL7, CCL8, CCL11, CCL13 and CXCL6) that attract and activate leukocytes. rs12075 is a missense variant and a *cis*-eQTL in whole blood for the *DARC* gene, which encodes the atypical chemokine receptor 1 (ACKR1) protein. STRINGdb analysis revealed that ACKR1 is an interacting partner for three (CCL2, CCL7 and CCL8) of the six *trans*-affected chemokines. Previous studies have shown that ACKR1 acts as a negative regulator of inflammation by nonspecifically binding both the CCL and the CXCL families of chemokines[19], suggesting an explanation for the multiple chemokine associations at this variant. Potentially downstream of its effects on chemokines, rs12075 is also associated with white blood cell count, as well as monocyte and basophil count[20] (Extended Data Figs. 5 and 6).

We found that plasma levels of some proteins were associated with numerous genetic loci, with IL-12B, KITL and TNFSF10 regulated by seven genetic loci each. We hypothesized that the mediating genes

at each of the regulatory loci for a given protein might be functionally related, enabling identification of shared pathways and/or the most likely mediating gene(s). We therefore generated protein–protein interaction networks for each of these multi-locus-regulated proteins and their respective candidate mediating genes (Extended Data Fig. 7). For TNFSF10, the network analysis linked genetic regulators of TNFSF10 to the plasminogen-activating system (Extended Data Fig. 7a and Supplementary Note 4). For KITLG, a driver of hematopoiesis[21], we found a cluster of interacting proteins, including PON1, ABCA1 and PLTP (Extended Data Fig. 7b) converging on cholesterol metabolism. Supporting this, we found that five of the seven *trans*-pQTLs for KITLG were significantly ($P \leq 5 \times 10^{-8}$, linear regression) associated with levels of either high-density lipoprotein- or low-density lipoprotein-cholesterol, and some with other lipids such as triglycerides and blood cell traits (Supplementary Table 10). These findings therefore suggest a link across plasma KITLG levels, cholesterol metabolism and altered hematopoiesis.

## Overlap with GWASs of traits and diseases
GWASs have identified thousands of genomic regions associated with common diseases[22], including immune-mediated diseases (IMDs).

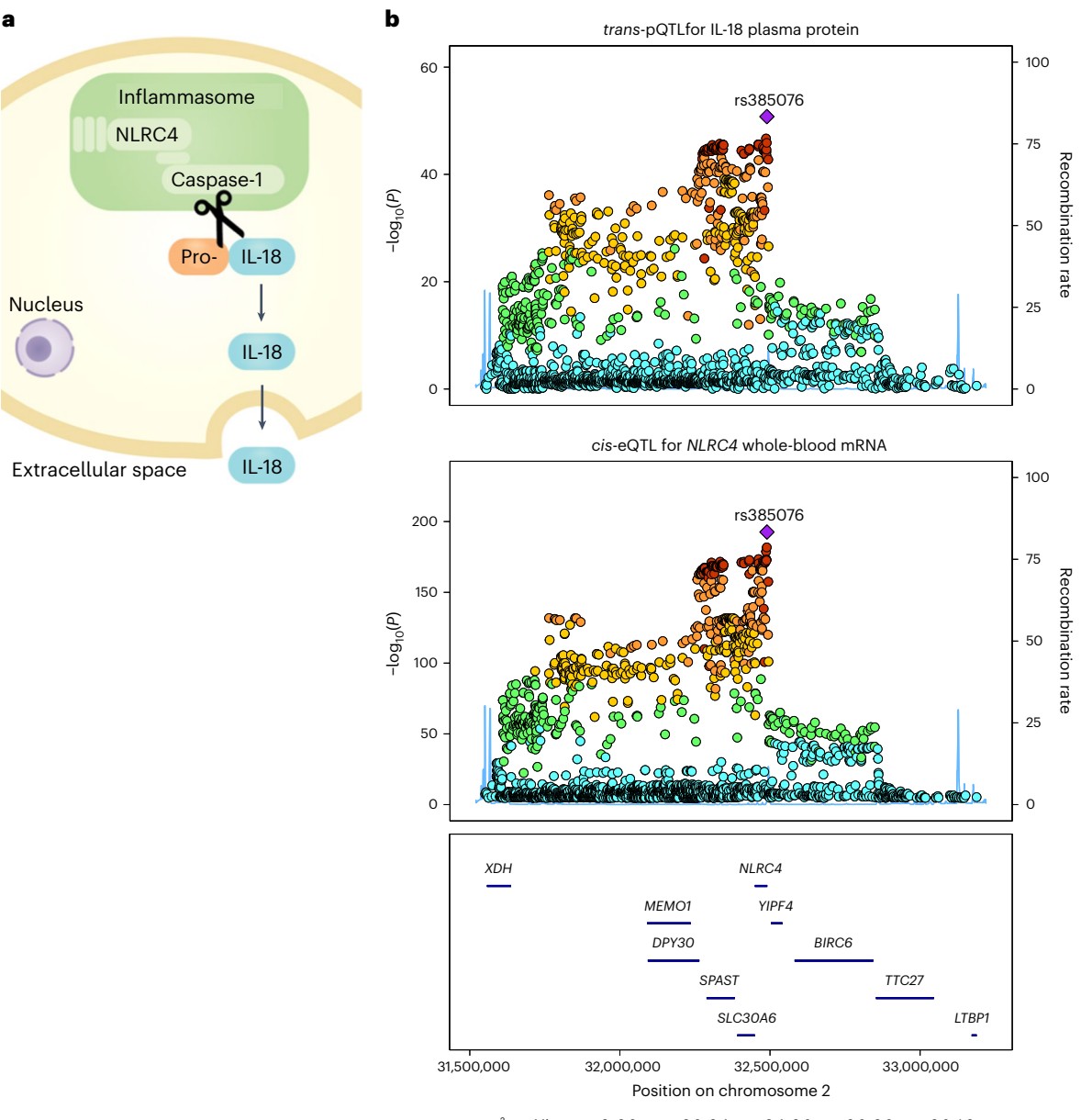

**Fig. 3 | Genetic regulation of the inflammasome affects plasma IL-18 levels.**
**a**, Schematic illustrating the cleavage of pro-IL-18 by caspase-1 and subsequent secretion of mature IL-18 from the cell into the extracellular space. **b**, Regional association plots around *NLRC4* showing: the *trans*-pQTL signal for plasma IL-18 protein (top) from the present study ($n$ = 14,824) and the *cis*-eQTL signal for *NLRC4* (bottom) in whole blood from the eQTLGen study ($n$ = 31,684)[14]. The purple diamond shows the sentinel pQTL variant. Other variants are colored by LD to the sentinel pQTL. Two-sided *P* values are from meta-analysis of linear regression estimates.

Many of these disease-associated loci lie outside protein-coding regions, leaving the effector molecules and pathways by which these genetic variants confer disease risk unclear. Integration of pQTL and GWAS data can help bridge this knowledge gap by linking disease risk loci to specific proteins. To this end, we looked for overlap between pQTLs, or proxy variants in high LD ($r^2 \geq 0.8$) with our sentinel variants, and disease-associated variants from GWASs. This revealed an overlap between our pQTLs and disease-associated variants for 73 diseases (Extended Data Fig. 8 and Supplementary Table 11). Examples of genetically anchored protein–disease connections included: TNFSF11 (RANKL) with osteoporosis and hypothyroidism, NGF (nerve growth factor) with migraine, TNFSF12 (TWEAK) with hypertension and fibroblast growth factor 5 (FGF5) with hypertension and cardiovascular diseases.

We next focused on IMDs in more detail, intersecting our pQTL data with IMD GWASs to identify proteins linking genotype and disease phenotypes. We found that 31 pQTLs overlap GWAS hits for at least 1 common IMD, with 76 unique pQTL protein–disease associations (Supplementary Table 12 and Extended Data Fig. 9). For example, we observed that a *cis*-pQTL for IL-10 was also associated with risk of inflammatory bowel disease (IBD), with the allele associated with higher plasma IL-10 correlating with reduced IBD risk, consistent with the anti-inflammatory effects of IL-10. Some pQTLs showed diverging directions of effect on different diseases; for example, the *trans*-pQTL at *IL6R* for plasma IL-6 levels described earlier had opposing directions of effect on risk of rheumatoid arthritis and allergic diseases (Extended Data Fig. 9), as previously described[23,24].

### *Trans*-pQTL implicates the *LTBR*–*LTA* axis in multiple sclerosis etiology

We identified a *trans*-pQTL for LTA (also known as TNF-β) at rs2364485 on chromosome 12 (Table 1), an intergenic variant previously found to be associated with multiple sclerosis[25]. We found that the multiple sclerosis risk allele, rs2364485:A, was associated with higher plasma levels of LTA. We next applied the ProGeM algorithm, which revealed two candidate genes in the region near the pQTL that might mediate the *trans*-pQTL: *TNFRSF1A* (encoding TNF receptor 1, TNFR1) and *LTBR* (encoding lymphotoxin β-receptor (LTBR)). LTA is a ligand for TNFR1, but can also bind the membrane-bound receptor LTBR when in complex with LTB. Functional studies have shown that *TNFRSF1A* is the causal gene underlying a neighboring, independent multiple sclerosis association in the region, about 70 kb upstream from rs2364485. The sentinel variant at this neighboring signal, rs1800693, results in an alternative *TNFRSF1A* isoform due to skipping of exon 6 (ref. 26). We therefore sought to determine whether *TNFRSF1A* is also the probable mediating gene for the LTA *trans*-pQTL at rs2364485, or whether *LTBR* is the more likely candidate. Through mining of eQTL databases, we found that rs2364485 is a *cis*-eQTL for *LTBR* (but not *TNFRSF1A*) in multiple tissues, including in the eQTLGen consortium meta-analysis of whole blood[14], with the multiple sclerosis risk allele (rs2364485:A) associated with reduced *LTBR* mRNA. Pairwise statistical colocalization analyses using conditioned *LTBR* eQTL data (from eQTLGen) and multiple sclerosis GWAS data[25] (Methods) showed that the rs2364485 *trans*-pQTL signal for LTA colocalizes with *LTBR* mRNA expression in both whole blood (PP = 0.79) and multiple sclerosis (PP = 0.86) (Fig. 4). Taken together, these data are consistent with a pathogenic model whereby the multiple sclerosis risk allele results in lower abundance of LTBR (the receptor) and consequently higher circulating levels of the ligand LTA.

### MR to identify protein drivers of IMDs

Observational studies comparing patients with IMDs with healthy controls have identified many proteins that are dysregulated. However, it is often unclear whether such proteins play causal roles in the disease process or are merely downstream markers. Distinguishing these possibilities is important therapeutically, because pharmacological targeting of the latter is unlikely to be beneficial. We therefore applied MR, an approach that tests the causal role of a risk factor ('exposure') in a disease in observational data using genetic variants as instrumental variables[27]. We used the 58 proteins with *cis*-pQTLs outside the human leukocyte antigen region in our study as exposures and 14 IMDs as outcomes (Methods). By restricting our use of genetic instruments to *cis*-pQTLs, we reduced the likelihood of violating MR assumptions through horizontal pleiotropy. Using generalized summary-data-based MR (GSMR)[28], we found 22 significant (false discovery rate (FDR) < 0.01) putative causal associations (Fig. 5 and Supplementary Table 13). To evaluate the robustness of these associations, we performed additional checks including evaluating the strength of the relevant disease association in the GWAS data and whether there might be confounding due to LD (Methods and Supplementary Table 14). After applying these filters, ten disease–protein pairs with robust evidence remained (Table 1). These results highlighted a number of established links between proteins and inflammatory diseases that are supported by other lines of evidence. For example, our finding that genetic predisposition to higher plasma levels of IL-12B (a subunit of IL-12) was associated with increased risk of IBD is consistent with the therapeutic benefit of ustekinumab (a monoclonal antibody targeting the p40 subunit common to IL-12 and IL-23) in IBD (Supplementary Table 15).

Our MR analysis implicated CXCL5, a chemokine that acts on neutrophils, in the etiology of UC. The plasma *cis*-pQTL for CXCL5 colocalized with *cis*-eQTLs for *CXCL5* in both blood and gut tissue and with the UC GWAS signal (Fig. 6a). To further explore the role of CXCL5 in UC, we compared expression of *CXCL5* transcripts in gut samples

## Table 1 | Putative causal protein–disease associations from MR analysis

| Protein | Disease | Odds ratio (95% CI) | P |
|---------|---------|---------------------|---|
| CD40 | Rheumatoid arthritis | 1.28 (1.21–1.37) | $1.4 \times 10^{-15}$ |
| CD40 | Multiple sclerosis | 0.75 (0.70–0.82) | $1.2 \times 10^{-12}$ |
| CD40 | Crohn's disease | 0.81 (0.75–0.87) | $2.2 \times 10^{-8}$ |
| CD40 | IBD | 0.87 (0.82–0.92) | $1.9 \times 10^{-6}$ |
| CD5 | Primary sclerosing cholangitis | 0.50 (0.35–0.70) | $8.1 \times 10^{-5}$ |
| CD6 | IBD | 1.10 (1.06–1.14) | $2.1 \times 10^{-7}$ |
| CXCL5 | UC | 0.79 (0.72–0.87) | $2.3 \times 10^{-6}$ |
| IL-12B | IBD | 1.38 (1.31–1.46) | $1.5 \times 10^{-30}$ |
| IL-12B | UC | 1.38 (1.29–1.48) | $4.7 \times 10^{-20}$ |
| IL-18R1 | Eczema | 1.15 (1.10–1.20) | $2.1 \times 10^{-10}$ |

IBD is based on GWASs in which Crohn's disease and UC cases are grouped together. *P* is the two-sided *P* value for the causal estimate of protein on disease from the GSMR package. The odds ratio (OR) is associated with a 1 s.d. increase in the protein level. OR > 1 indicates that genetic propensity to higher levels of the plasma protein is associated with higher disease risk and OR < 1 with reduced risk. CI, confidence interval.

from patients with IBD and healthy controls using the IBD Transcriptome and Metatranscriptome Meta-Analysis (IBD TaMMA) platform[29]. We observed that *CXCL5* gene expression was significantly increased in mucosal biopsies from patients with UC compared with biopsies from healthy control participants ($\log_2$(fold-change) ($\log_2$(FC)) = 7.07, $P = 1.98 \times 10^{-174}$, Wald test) (Fig. 6b). Indeed, *CXCL5* was the third most highly upregulated transcript across the transcriptome (Fig. 6c). We replicated these findings in three independent datasets (Fig. 6d). Of note, our MR analysis revealed that the association of CXCL5 was restricted to UC (unadjusted $P = 2.3 \times 10^{-6}$, GSMR), with no significant association in Crohn's disease (CD; unadjusted $P = 0.4$) (Fig. 6a,e). Supporting this specific pathogenic effect, *CXCL5* gene expression in gut samples from patients with IBD was higher in UC than in CD (Fig. 6b). Counterintuitively (given the upregulation of *CXCL5* in tissue samples of patients with UC), evaluation of the direction of MR association effect revealed that genetic susceptibility to higher plasma CXCL5 reduces the risk of UC (Fig. 6e). This effect was consistent across 12 of the 13 individual genetic variants used in the MR score (Extended Data Fig. 10a). We found consistent directions of effect for the CXCL5 plasma pQTLs and the blood and gut eQTLs (Extended Data Fig. 10b), indicating that our results are generalizable at both the mRNA and protein levels and across local and systemic sites. Together, these data indicate that genetic tendency to lower CXCL5 is a causal risk factor for development of UC, despite the strong upregulation of CXCL5 once disease develops.

We observed that genetic predisposition to higher plasma CD40 levels was associated with increased rheumatoid arthritis risk, consistent with evidence from both animal models and humans implicating the CD40 pathway in rheumatoid pathogenesis[30]. In addition, our MR analysis identified a potential causal role for the CD40 pathway in IBD (including both CD and UC) and multiple sclerosis. However, the MR associations for these diseases had the opposite direction of effect compared with rheumatoid arthritis; that is, genetic predisposition to lower plasma CD40 levels was associated with higher risk of IBD and multiple sclerosis. These findings highlight how the same pathway can

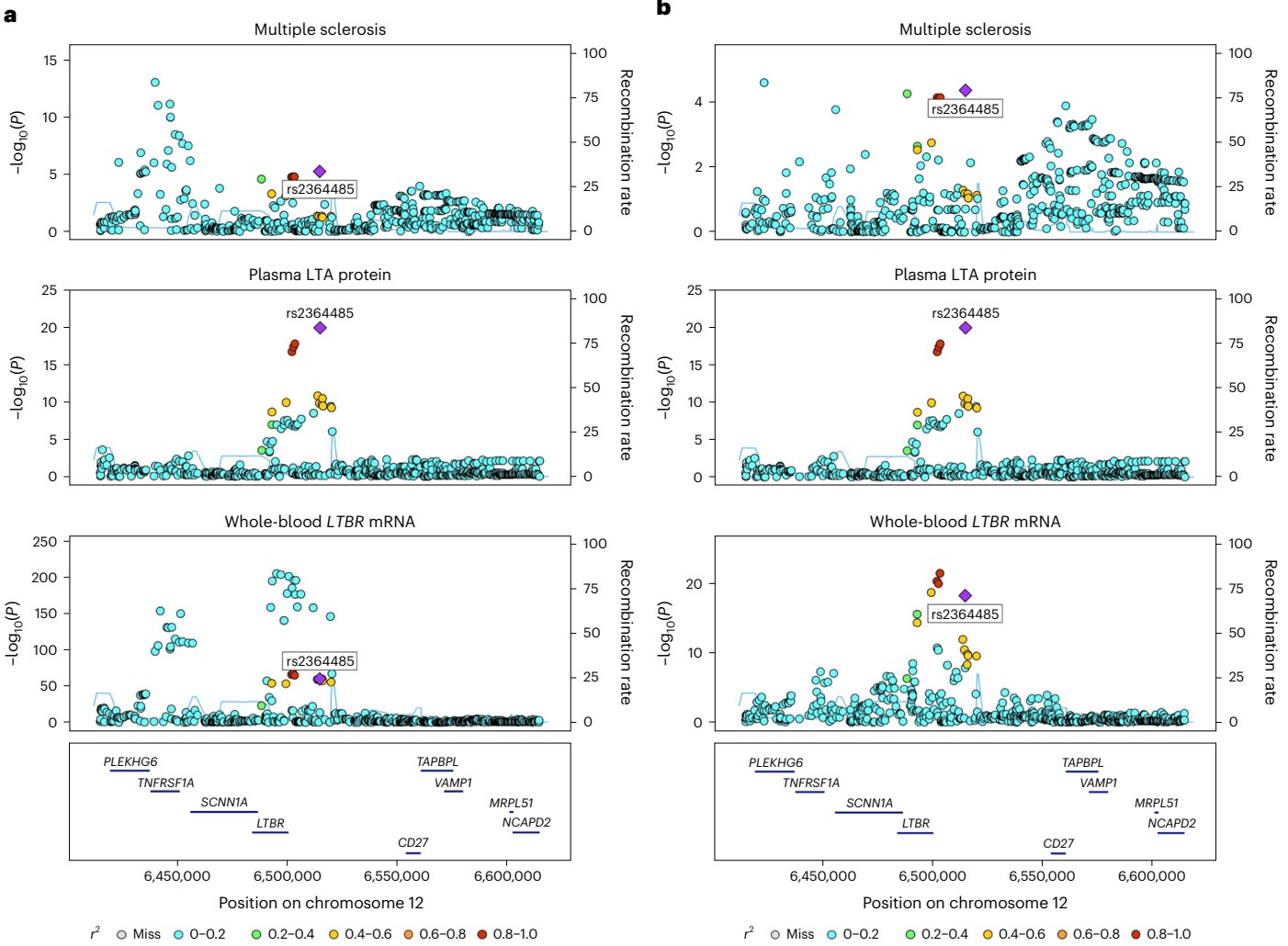

**Fig. 4 | The *LTBR–LTA* axis in the etiology of multiple sclerosis.**
**a**,**b**, Unconditioned (**a**) and conditioned (**b**) regional association plots at the
*TNFRSF1A-LTBR* locus (rs2364485 ± 100 kb) for multiple sclerosis (top), plasma
LTA protein levels (middle) and *LTBR* mRNA expression in whole blood from
eQTLGen[14] (bottom). Multiple sclerosis associations were conditioned on
rs1800693 (the strongest disease signal in the region). *LTBR* mRNA expression

levels were conditioned on the following independent eQTLs: rs3759322,
rs1800692, rs2228576, rs10849448, rs2364480 and rs12319859. The purple
diamond shows the sentinel pQTL variant. Other variants are colored by LD
to the sentinel pQTL. Two-sided *P* values are from meta-analysis of linear
regression estimates.

have pleiotropic effects on disease susceptibility, but also point to the
complexity of immune-mediated disease pathogenesis, with opposing
effects on different diseases.

## Discussion

In the present study, we performed a large-scale pQTL GWAS of 91 cir-
culating inflammation-related proteins measured using Olink immu-
noassays, identifying 180 significant primary pQTL signals (59 *cis*, 121
*trans*). Colocalization analysis suggested that only a small proportion
of the plasma *cis*-pQTLs reported in the present study are underpinned
by the same causal genetic variant as the whole-blood *cis*-eQTL for the
corresponding gene. Of note, the plasma proteome is not the direct cor-
ollary of the whole-blood transcriptome: plasma pQTL studies examine
genetic effects on extracellular protein levels, whereas blood eQTL
studies examine the effects on intracellular RNA levels (predominantly
in leukocytes). This has several implications. First, plasma protein levels
can be affected by nontranscriptional mechanisms including cleavage,
secretion and clearance. Second, a wide range of tissues other than
blood cells (for example, the liver) contribute to the plasma proteome.
This is evident when considering circulating proteins that are measured

as biomarkers in clinical practice (for example, albumin produced by
the liver, troponin by the heart, prostate-specific antigen by the pros-
tate). Indeed, by extending our comparison across multi-tissue eQTL
databases, we showed that at least 50% of the *cis*-pQTLs we observed
are probably driven by cognate *cis*-eQTLs in a diverse range of tissues
and cell types. Blood eQTL studies have been carried out using sample
sizes similar to the sample size in our pQTL study. The eQTL studies
in other tissues tend to be smaller and so it is likely that some of the
plasma *cis*-pQTLs observed in the present study are underpinned by
tissue-specific eQTLs that have not yet been detected due to lack of
statistical power. Finally, other mechanisms such as alternative splic-
ing might account for some *cis*-pQTLs without corresponding eQTLs.

Our pQTL study identified twice as many *trans* associations
compared with *cis* (121 versus 59, respectively), in keeping with other
well-powered pQTL studies (for example, refs. 7–9). The integration
of *cis*-pQTLs (and *cis*-eQTLs) with GWAS data provides useful, if some-
times obvious, insights into the upstream mechanisms of disease,
because the mediating gene has usually already been suspected by
virtue of the location of the GWAS signal. In contrast, *trans*-pQTLs rep-
resent a double-edged sword for interpreting genetic associations with

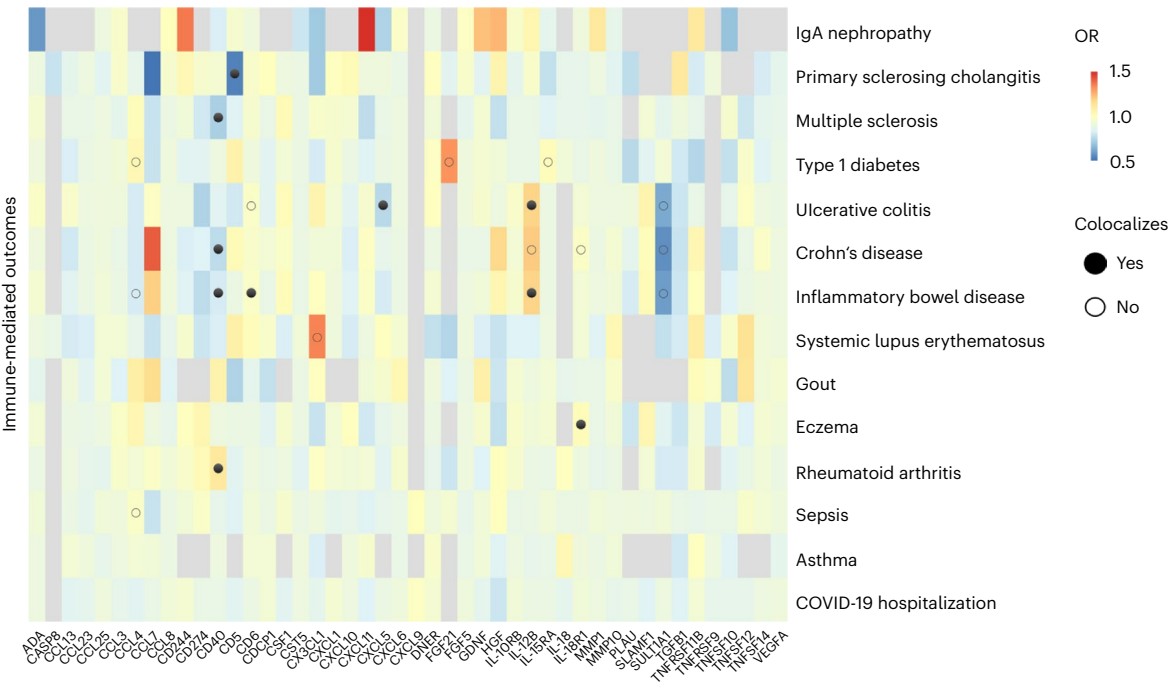

**Fig. 5 | MR analysis of circulating proteins in immune-mediated disease etiology.** GSMR analysis[28] using *cis*-pQTLs as genetic instruments to test the causal role of plasma proteins across IMDs. Cells are colored according to the effect size and direction: red indicates that higher genetically predicted plasma protein levels are associated with increased disease risk; blue indicates that higher genetically predicted plasma protein levels are associated with reduced disease risk; and gray represents no result because fewer than three variants were available for GSMR analysis. Associations with FDR ≤ 0.01 are denoted with dots, with filled circles indicating those that were robust to confounding by LD and open circles indicating those that were not.

disease. On the one hand, they often represent a less direct link from genotype to disease than *cis*-pQTLs and, from the perspective of causal inference analysis, are more vulnerable to violating the assumptions of MR through horizontal pleiotropy. On the other hand, they can reveal molecular mediators of disease encoded by genes distant from the disease GWAS signal. For example, we identified a *trans*-pQTL (rs2364485) for LTA at a multiple sclerosis risk locus. This multiple sclerosis risk locus contains two plausible causal genes (*TNFRSF1A* and *LTBR*) and two independent signals for multiple sclerosis risk (rs1800693 and rs2364485). By integrating whole-blood eQTL and multiple sclerosis GWAS data, we showed that *LTBR* is the most likely gene mediating the LTA *trans*-pQTL at rs2364485, and one of the multiple sclerosis signals at the locus. LTA is a member of the TNF superfamily of proteins and is the only member of this superfamily that is generated as a secreted protein rather than through cleavage of a membrane-bound protein. The multiple sclerosis risk allele is associated with lower expression of *LTBR* and higher circulating protein levels of LTA, a component of its ligand. This raises the question of whether elevated LTA is secondary to lower LTBR, or vice versa (for example, through compensatory receptor downregulation). The distinction between *cis*- and *trans*-QTLs enables us to address this. Given that the eQTL for *LTBR* is *cis* and the pQTL for LTA is *trans*, it is highly likely that the former is the upstream effect, with the higher levels of soluble LTA occurring as a result of reduced binding to its receptor. This demonstrates the value of pairing QTLs for ligands and their receptors for deconvoluting the ordering of biological pathways.

Integration of pQTLs with GWAS disease signals revealed disease–protein connections reflecting both established and plausible putative mechanisms of pathophysiology. For example, a *cis*-pQTL for TNFSF11 (RANKL) overlapped with GWAS signals for osteoporosis and hypothyroidism. The former is consistent with RANKL's well-established role in bone biology and RANKL is the target of the anti-osteoporosis drug denosumab[31]. However, RANKL also plays a role in the immune system[32] and these effects may be relevant to risk of autoimmune hypothyroidism. A *cis*-pQTL for TNFSF12 (TWEAK) was associated with risk of hypertension. TWEAK is a cytokine predominantly produced by leukocytes and has pleiotropic actions, including on the endothelium[33,34], potentially explaining the association with blood pressure. A *cis*-pQTL for FGF5 was also associated with susceptibility to hypertension and cardiovascular diseases, with the allele associated with higher plasma FGF5 levels being associated with lower risk of cardiovascular diseases. Consistent with this, there are reports that FGF5 has cardioprotective effects in pig models[35].

Of our pQTLs, 31 overlap GWAS hits for at least one common IMD. Disease–protein links identified from this analysis highlighted commonalities in pathogenesis between specific IMDs, mirroring the overlap in clinical manifestations. However, the contributions of proteins to IMD risk were sometimes complex, with the same protein conferring risk of one IMD but protecting from another. For example, genetic predisposition to higher levels of soluble IL-6 had opposing effects on risk of rheumatoid arthritis and allergic disease. We observed a similar phenomenon for CD40, with genetic predisposition to higher CD40 increasing risk of rheumatoid arthritis but protecting against IBD and multiple sclerosis.

The development of biologic therapies targeting specific inflammatory proteins has transformed the clinical management of immune-mediated diseases[36]. Understanding which proteins are drivers of disease and distinguishing these from proteins that are simply markers of inflammation is therefore important for the development of new treatments. To this end, we used MR to evaluate the causal contributions of proteins to different IMDs. Our results identify pathways that are already the target of existing drugs (for example, IL-12B in IBD), providing confirmation of the utility of this approach, and also highlight new potential therapeutic targets.

One such example was the CD40 pathway in rheumatoid arthritis. CD40 is a stimulatory receptor constitutively or inducibly expressed on

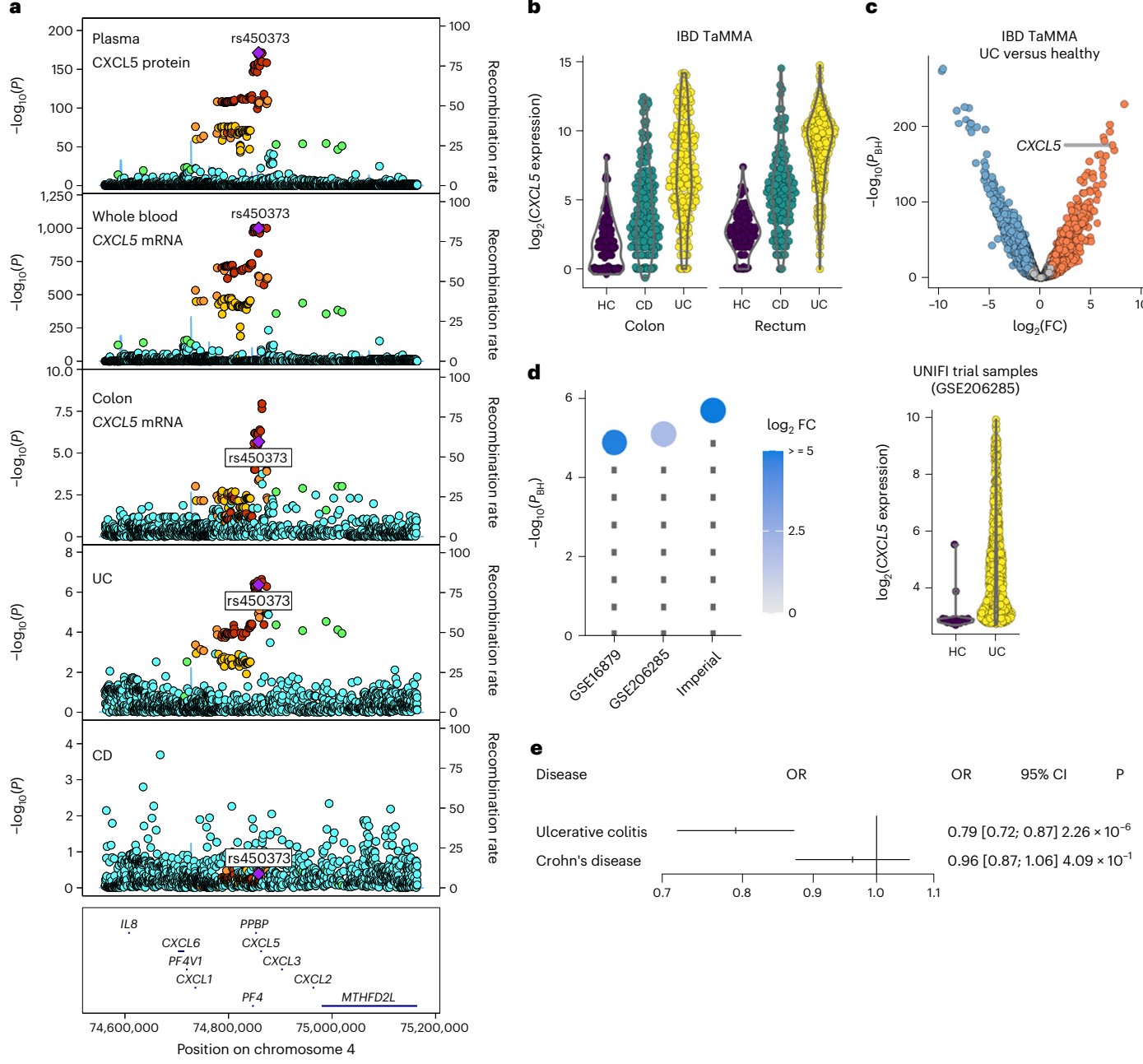

**Fig. 6 | CXCL5 in UC pathogenesis. a**, Genetic associations in the *CXCL5* gene region. From top to bottom: plasma CXCL5 pQTL (*n* = 14,824 participants), whole-blood eQTL (from eQTLGen data, *n* = 31,684 participants), colon tissue eQTL (GTEx, *n* = 368 individuals), UC (cases = 12,366, controls = 33,609) and CD (cases = 12,194, controls = 28,072) (from the IBD Genetics Consortium[51]). The purple diamond shows the sentinel pQTL variant. Other variants are colored by LD to the sentinel pQTL. *P* values are from linear regression for QTLs and logistic regression for case-control GWAS. **b**, Violin plots showing *CXCL5* expression in gut mucosal samples from patients with UC or CD and healthy controls (HC) in IBD TaMMA. **c**, Volcano plot showing differential expression analysis comparing colonic tissue from UC with HCs (IBD TaMMA). Red and blue points represent significantly (5% FDR) up- and downregulated transcripts, respectively. Gray indicates nonsignificant. *P*_BH, Benjamini–Hochberg adjusted *P* values. *P* values in **c** and **d** are from Wald tests (two sided). **d**, Replication. Left, *CXCL5* differential expression in colon biopsies in UC versus HCs from transcriptome-wide analysis across three cohorts. The GSE numbers are GEO accession numbers. Imperial is the Imperial UC cohort. Each lollipop represents a separate cohort: GSE16879 (*n* = 24 UC patients versus *n* = 6 HCs); GSE206285 (*n* = 550 UC patients versus *n* = 18 HCs); and Imperial (*n* = 16 UC versus 6 HCs). The circle color indicates the log₂(FC) in *CXCL5* expression between UC and HCs. Right, *CXCL5* expression in colon biopsies sampled at baseline during the UNIFI clinical trial. Each point represents an individual. **e**, Forest plot showing MR analysis for UC and CD. OR is the odds ratio for the risk associated with a 1 s.d. increase in the level of the protein. The center of the bar is the point estimate for OR and the whiskers are the 95% CIs.

both immune and nonimmune cells[37]. Its ligand, CD40L, is expressed primarily on activated T cells but also on a range of other cell types. CD40L–CD40 binding triggers immune cell activation and proliferation and inflammatory cytokine production and the differentiation of B cells into immunoglobulin (Ig)G-secreting plasma cells, making it central to antibody responses. In a murine model of inflammatory arthritis, knock-out or inhibition of the CD40 pathway resulted in reduced inflammation[38]. Observational studies have demonstrated upregulation of CD40L in the blood and tissues of patients with rheumatoid arthritis and other autoimmune rheumatic diseases[30,39].

These findings motivated development of drugs targeting the CD40 pathway in rheumatoid arthritis and other IMDs, but anti-CD40L therapy was complicated by thrombosis due to cross-linking CD40L on platelets. Therapeutic targeting of CD40 rather than CD40L may avoid this. Our MR results suggest rheumatoid arthritis as a candidate for this approach. However, the directionally discordant effects we observed of CD40 on rheumatoid arthritis versus multiple sclerosis and IBD raises the possibility of triggering other forms of immune-mediated diseases as a side effect of anti-CD40 therapy. This has some parallels with therapies targeting TNF, which are effective in rheumatoid arthritis but not in multiple sclerosis, and indeed can worsen multiple sclerosis or provoke de novo central nervous system demyelination[40,41].

Our MR findings implicate CXCL5 in the etiology of UC, where genetic susceptibility to higher levels of plasma CXCL5 was associated with lower UC risk. Examination of eQTL data revealed that this observation was consistent at the RNA level in both blood and gut tissue. By contrast, in our case–control analysis comparing gut tissue from patients with UC with that from controls, *CXCL5* is one of the most upregulated transcripts. A previous study reported that serum levels of CXCL5 are higher in IBD patients than in controls[42]. Recent studies using UC gut tissue have implicated upregulation of genes encoding neutrophil-targeting chemokines, including *CXCL5*, by nonimmune cells as correlating with important histopathological features, such as ulceration, and differentiating patient trajectories, including their responsiveness to different treatments[43,44]. Targeting CXCR2, the receptor for CXCL5, significantly attenuates animal models of UC[44]. One possible explanation that may reconcile these apparently contradictory findings is that genetic tendency to lower CXCL5 expression increases UC risk through impaired mucosal immune homeostasis, but that elevated CXCL5 is an important driver of tissue injury once disease has been initiated. By analogy, a noncoding genetic variant associated with lower gene and protein expression of *TNFSF15* (encoding the inflammatory cytokine TL1A) in monocytes and macrophages increases IBD susceptibility[45], but TL1A is upregulated both systemically and in the gut in patients with active IBD[46,47], and anti-TL1A therapies have recently shown efficacy in IBD in phase 2 randomized trials (NCT05013905 and NCT04996797 (ref. [48])).

Our study has several limitations. Our pQTL analysis was restricted to 91 proteins, limiting the generalizability of our findings, particularly with regard to genetic architecture. As this was a pQTL meta-analysis, study-level technical variation resulted in heterogeneity, which necessitated the filtering out of potentially spurious associations that were inconsistent across cohorts. There is a risk that some true biological signals were also removed in this process. Very large single cohorts with standardized sample processing such as UK Biobank will avoid this issue. Our meta-analysis consisted predominantly of general population cohorts without inflammatory disease. There may be context-specific pQTLs that are present only during infection or inflammation, which our study may not have detected. By analogy, eQTL studies using human immune cells stimulated in vitro (for example, with lipopolysaccharide or interferon) have demonstrated eQTLs that are not present in resting cells but become apparent in the context of cellular activation[49,50]. Conducting well-powered pQTL studies in patients with inflammation will be an important future research endeavor. Where proteins exist in both membrane-bound and cleaved states, it is not always clear whether plasma proteomic assays are exclusively capturing the soluble form or also protein from cell membranes (for example, arising from in vivo sources such as exo-/ectosomes or ex vivo processes such as venepuncture or sample processing). This complicates the interpretation of the direction of effect from MR analysis. Future well-powered studies examining genetic determinants of cell-surface protein expression measured through flow cytometry would provide valuable complementary information to aid the interpretation of plasma pQTL studies. Finally, as with all epidemiological-scale pQTL studies, proteins were measured in plasma (that is, the extracellular component of blood), which may not always be the disease-relevant biological compartment, and where the direction of genotype-expression association may even be opposite to the site of inflammation. Thus, future tissue- and cell-specific pQTL studies will be valuable to understand differences in genetic signals across tissues.

In summary, we have used a large international consortium to identify the genetic determinants of a set of inflammation-related proteins, providing insight into the etiology of immune-mediated diseases. The pQTL summary statistics generated in the present study will be a valuable resource for interrogating future disease GWASs and guiding drug target identification and prioritization.

## Online content

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

Jing Hua Zhao[1,2,36], David Stacey[1,2,3,4,36], Niclas Eriksson [5], Erin Macdonald-Dunlop [6], Åsa K. Hedman[7,8], Anette Kalnapenkis[1,9], Stefan Enroth [10], Domenico Cozzetto [11], Jonathan Digby-Bell[12], Jonathan Marten[1], Lasse Folkersen [13], Christian Herder [14,15,16], Lina Jonsson[17], Sarah E. Bergen[7], Christian Gieger [18,19], Elise J. Needham[1,2], Praveen Surendran[1,20,21], Estonian Biobank Research Team*, Dirk S. Paul [1,2,20,22], Ozren Polasek[23], Barbara Thorand [16,18], Harald Grallert[16,18,19], Michael Roden [14,15,16], Urmo Võsa[9], Tonu Esko[9], Caroline Hayward [24], Åsa Johansson [10], Ulf Gyllensten [10], Nick Powell[11], Oskar Hansson [25,26], Niklas Mattsson-Carlgren[27,28,29], Peter K. Joshi [6], John Danesh[1,2,20,30,31], Leonid Padyukov [32,33], Lars Klareskog [32,33], Mikael Landén [7,17], James F. Wilson [6,24], Agneta Siegbahn[34], Lars Wallentin[34], Anders Mälarstig [7,8], Adam S. Butterworth [1,2,20,21,30,37] ✉ & James E. Peters [1,21,35,37] ✉

[1]British Heart Foundation Cardiovascular Epidemiology Unit, Department of Public Health and Primary Care, University of Cambridge, Cambridge, UK. [2]Victor Phillip Dahdaleh Heart and Lung Research Institute, University of Cambridge, Cambridge, UK. [3]Australian Centre for Precision Health, Unit of Clinical and Health Sciences, University of South Australia, Adelaide, South Australia, Australia. [4]South Australian Health and Medical Research Institute, Adelaide, South Australia, Australia. [5]Uppsala Clinical Research Center, Uppsala University, Uppsala, Sweden. [6]Centre for Global Health Research, Usher Institute, University of Edinburgh, Edinburgh, UK. [7]Department of Medical Epidemiology and Biostatistics, Karolinska Institutet, Stockholm, Sweden. [8]Development and Medical, Pfizer Worldwide Research, Stockholm, Sweden. [9]Estonian Genome Center, Institute of Genomics, University of Tartu, Tartu, Estonia. [10]Department of Immunology, Genetics, and Pathology, Biomedical Center, SciLifeLab Uppsala, Uppsala University, Uppsala, Sweden. [11]Department of Metabolism, Digestion and Reproduction, Faculty of Medicine, Imperial College London, London, UK. [12]School of Immunology and Microbial Sciences, King's College London, London, UK. [13]Nucleus Genomics ltd, New York, NY, USA. [14]Institute for Clinical Diabetology, German Diabetes Center, Düsseldorf, Germany. [15]Department of Endocrinology and Diabetology, Medical Faculty and University Hospital Düsseldorf, Heinrich Heine University Düsseldorf, Düsseldorf, Germany. [16]German Center for Diabetes Research, Munich-Neuherberg, Germany. [17]Institute of Neuroscience and Physiology, University of Gothenburg, Gothenburg, Sweden. [18]Institute of Epidemiology, Helmholtz Zentrum München, German Research Center for Environmental Health, Neuherberg, Germany. [19]Research Unit of Molecular Epidemiology, Helmholtz Zentrum München, German Research Center for Environmental Health, Neuherberg, Germany. [20]British Heart Foundation Centre of Research Excellence, School of Clinical Medicine, Addenbrooke's Hospital, University of Cambridge, Cambridge, UK. [21]Health Data Research UK, Wellcome Genome Campus and University of Cambridge, Hinxton, UK. [22]Centre for Genomics Research, Discovery Sciences, BioPharmaceuticals R&D, AstraZeneca, Cambridge, UK. [23]Medical School, University of Split, Split, Croatia. [24]MRC Human Genetics Unit, Institute of Genetics and Cancer, University of Edinburgh, Western General Hospital, Edinburgh, UK. [25]Clinical Memory Research Unit, Department of Clinical Sciences Malmö, Lund University, Lund, Sweden. [26]Skåne University Hospital, Malmö, Sweden. [27]Wallenberg Centre for Molecular Medicine, Lund University, Lund, Sweden. [28]Clinical Memory Research Unit, Faculty of Medicine, Lund University, Lund, Sweden. [29]Department of Neurology, Skåne University Hospital, Lund University, Lund, Sweden. [30]NIHR Blood and Transplant Research Unit in Donor Health and Behaviour, University of Cambridge, Cambridge, UK. [31]Department of Human Genetics, Wellcome Sanger Institute, Hinxton, UK. [32]Division of Rheumatology, Department of Medicine (Solna), Karolinska Institutet and Karolinska University Hospital, Stockholm, Sweden. [33]Center for Molecular Medicine, Karolinska Institutet, Stockholm, Sweden. [34]Department of Medical Sciences and Uppsala Clinical Research Center, Uppsala University, Uppsala, Sweden. [35]Department of Immunology and Inflammation, Imperial College London, London, UK. [36]These authors contributed equally: Jing Hua Zhao, David Stacey. [37]These authors jointly supervised this work: Adam S. Butterworth, James E. Peters. *A list of authors and their affiliations appears at the end of the paper. ✉e-mail: asb38@medschl.cam.ac.uk; j.peters@imperial.ac.uk

## Estonian Biobank Research Team

Andres Metspalu[9], Lili Milani[9], Reedik Mägi[9], Mari Nelis[9] & Georgi Hudjašov[9]

## Methods

### Cohorts
We recruited 11 cohorts, totaling 14,824 participants, with genome-wide genetic data and plasma proteomic data measured using the Olink Target Inflammation panel. All participants provided written, informed consent. No statistical methods were used to predetermine sample sizes but our sample sizes are similar to or larger than those reported in previous publications[1–4,7–9]. Cohort details are provided in Supplementary Note 1.

### Protein assays
Plasma proteins were measured using the Olink Target-96 Inflammation immunoassay panel, which measures 92 inflammation-related proteins. Proteomic data for each cohort were generated at Olink laboratories in Uppsala. During the course of the project, brain-derived neurotrophic factor (BDNF) was removed from the inflammation panel by Olink due to assay problems, so 91 proteins were included in our study (Supplementary Table 2). Normalized Protein eXpression (NPX) is Olink's normalized relative units on a log₂ scale. Olink defines the LLOD for quantification of each protein as 3 s.d. above background (determined using blank control samples), but provides NPX as continuous data which can include values below the calculated LLOD. We had access to individual-level data for INTERVAL, the largest contributing cohort ($n = 4,896$) and used this to calculate the proportion of samples less than the LLOD for each protein (Extended Data Fig. 2a).

### Genotyping
Each cohort was genotyped on an SNP array and imputed using either a 1000 Genomes or Haplotype Reference Consortium (HRC) panel (Supplementary Table 1).

### Cohort-level pQTL mapping
In each cohort, a GWAS analysis was run for each protein using linear regression (additive genetic association model) with protein level as the dependent variable. Proteins were inverse-rank normalized before linear regression and thus met the assumptions of the statistical test. Population substructure was adjusted for by including genetic principal components as covariates. We also included age, sex and other study-specific covariates in the model (Supplementary Table 1). To avoid proteins with truncated distributions due to LLOD, with multiple tied values that would violate linear regression assumptions, pQTL analysis was performed using continuous protein values (including those below the LLOD where relevant). We illustrate the value of this approach in recovering biological signals in Extended Data Fig. 2b.

### The pQTL meta-analysis
We meta-analyzed pQTL summary statistics from each cohort (Supplementary Table 1), representing a total of 14,824 participants. A schematic of our analysis pipeline is shown in Extended Data Fig. 1. Before the meta-analysis, we applied cohort-level filters to pQTL GWAS summary statistics with respect to MAF (≥0.001), Hardy–Weinberg equilibrium ($P > 10^{-6}$) and imputation score ($r^2 \geq 0.3$ or SNPTEST proper_info≥0.4). For each cohort, we generated QQ plots and Manhattan plots for visual examination using the R packages qqman v.0.1.4 and QCG-WAS v.1.0-8. We performed the fixed-effects meta-analysis using the METAL software (v.28.8.2018), and inverse-variance weighted analysis of regression betas and standard errors from the cohort-level summary statistics. From the meta-analysis summary statistics, we calculated the genomic inflation factor for each protein GWAS and generated QQ and Manhattan plots (Supplementary Fig. 1). We generated Forest plots to examine intercohort heterogeneity using the gap package v.1.2.3-6. Regional association plots were generated using LocusZoom 1.4 (Supplementary Fig. 2). We defined statistical significance as $P \leq 5 \times 10^{-10}$ (based on Bonferroni correction of the conventional 'genome-wide' significance threshold $P \leq 5 \times 10^{-8}$ for approximately 100 proteins).

To remove potentially erroneous meta-analysis signals arising due to a strong association in a single cohort, we examined the meta-analysis results at each sentinel variant by visual inspection of the Forest plot and imposed the following criteria: (1) to be included in the meta-analysis, a variant was required to be present in at least 3 studies and at least 3,500 participants; and (2) to be declared significant, we required a meta-analysis $P \leq 5 \times 10^{-10}$ and, if there was evidence of heterogeneity with $I^2 > 30\%$, then we required the $P$ value in at least three studies to be <0.05 and the direction of effect in those studies to be consistent with the overall meta-analysis results. These were implemented through modification of the METAL source code.

### Replication cohort
We compared the results from our primary meta-analysis with pQTL results generated in an independent set of 1,585 participants from the ARISTOTLE study[12,52].

### Definition of pQTL sentinel variants and regions
We defined a pQTL as a genetic locus significantly ($P \leq 5 \times 10^{-10}$) associated with protein abundance. We defined the sentinel variant at a locus as the variant with the lowest $P$ value in the region for a given protein. We used the following approach for each protein to define genomic regions and the sentinel variant in each: (1) we first obtained a list of significant ($P \leq 5 \times 10^{-10}$) variants and the flanking region (±1 Mb) for each variant; (2) overlapping regions were then iteratively merged until no overlapping regions remained; and (3) the most significant variant in each resulting region was then defined as the sentinel variant. This approach has the flexibility to cope with long stretches of LD while avoiding the drawback of setting a longer than necessary region for all variants. The algorithm was implemented using bedtools v.2.27.0. Signals within 1 Mb of the transcription start site (TSS) of the gene encoding the target protein were defined as *cis* and those beyond 1 Mb as *trans*.

### Protein variance explained by pQTLs
We used the following equation to estimate the proportion of variance explained (PVE) by ($T$) pQTLs from the meta-analysis summary statistics for each protein:

$$\text{PVE} = \sum_{i=1}^{T} \frac{\chi_i^2}{\chi_i^2 + N_i - 2} \tag{1}$$

where $\chi_i^2$ is the $\chi^2$ score for pQTL variant $i$ calculated from its estimated effect size and standard error and $N_i$ is the associated sample size.

### Conditional analysis
To identify conditionally independent signals within a genomic region, we performed approximate stepwise conditional analyses using GCTA v.1.93.0beta with the '--cojo-slct' option, using estimated effect sizes and standard error values from the meta-analysis. We estimated the correlation between variants using individual-level data from the INTERVAL study. As GCTA imputes LD from mean genotypes when they are missing, to avoid bias we excluded variants with MAF < 1% (unless they were sentinel variants). For stepwise selection, we considered only those variants passing the genome-wide threshold ($P \leq 5 \times 10^{-10}$), rather than all variants in the region. As in certain cases GCTA conditional analysis yielded results involving pairs of variants in relatively high LD ($r^2 \geq 0.7$), we restricted the results to independent genetic variants (defined as $r^2 \leq 0.1$ (ref. 53), based on LD calculation in the INTERVAL cohort, where we had access to individual-level genotype data) while forcing the inclusion of the sentinel variants in the pruned set[54] (Supplementary Table 4).

### Identification of known pQTLs
To identify previously reported pQTLs, we manually curated published results from the literature obtained from the National Center

for Biotechnology Information's (NCBI's) web interface (https://pubmed.ncbi.nlm.nih.gov) through its Entrez programming utility R/rentrez[55], PhenoScanner v.2 (ref. 56) and the NHGRI-EBI GWAS catalog with phenotypes mapped to the experimental factor ontology (EFO) EFO_0004747 (protein measurement), restricting the results to associations reported in European-ancestry populations. We considered previously reported pQTLs to be variants that reached the conventional genome-wide significance threshold $P \leq 5 \times 10^{-8}$ and that were in high LD ($r^2 \geq 0.8$) with the pQTL sentinel variant from our meta-analysis.

### Variant annotation

We obtained the absolute distance of sentinel variants to the TSS of the gene encoding the target protein using the rGREAT (Genomic Regions Enrichment of Annotations Tool)[57] R package. We annotated sentinel variants and LD proxies (defined as $r^2 \geq 0.8$, using the INTERVAL dataset as the LD reference panel) and Ensembl's Variant Effect Predictor (VEP, v.98.3) including the LOFTEE plugin.

### The eQTL–pQTL colocalization analysis

We performed pairwise statistical colocalization analyses of cis-pQTLs identified in the meta-analysis with cognate cis-eQTL data from eQTLGen[14], the eQTL Catalogue[17] and GTEx v.8 (ref. 15). We extracted the meta-analysis summary statistics for each cis-pQTL sentinel and their ±1 Mb flanking regions, then extracted the same genomic windows from their cognate cis-eQTL data. eQTLGen comprises eQTL data from 31,684 participants on 19,250 genes that are robustly expressed in blood (https://www.eqtlgen.org/cis-eqtls.html). Of our 59 cis-pQTLs, there was genome-wide significant ($P \leq 5 \times 10^{-8}$) cis-eQTL for 40 genes in the eQTLGen data. One gene (*TGFB1*) had a cis-eQTL at FDR < 0.05 but that was not genome-wide significant ($P = 1.8 \times 10^{-7}$) and two had no eQTL association (*IL17C*, *TNFSF11*). Sixteen genes had no eQTL data in the eQTLGen summary statistics, presumably due to lack of robust expression in blood; these were: *CCL11*, *CCL13*, *CCL19*, *CCL20*, *CCL7*, *CST5*, *CX3CL1*, *CXCL11*, *DNER*, *FGF21*, *FGF5*, *GDNF*, *IL12B*, *MMP10*, *NGF* and *TNFRSF11B*.

For GTEX v.8 and the eQTL Catalogue, all 59 cis-pQTLs had corresponding eQTL summary statistics available for colocalization testing across one or more tissues. We performed colocalization analyses using the coloc R package as implemented in v.5.2.2 of the eQTL Catalogue/colocalization workflow[17] (https://github.com/kauralasoo/eQTL-Catalogue-resources). Coloc returns posterior probabilities indicating the likelihood that the following scenarios are true: there is no association at the locus with either protein or mRNA (PP0); there is an association with protein abundance but not mRNA expression (PP1); there is no association with protein abundance but there is an association with mRNA expression (PP2;) there is an association with both the protein and the mRNA but with distinct causal variants (PP3); there is an association with both the protein and the mRNA with a shared causal variant (PP4). We considered a PP4 ≥ 0.8 to be robust evidence of colocalization between a cis-pQTL and its cognate cis-eQTL. As eQTLGen data only provide allele frequency (*f*) and *z*-score statistic for a particular variant, we obtained the effect size (*b*) and its standard error (s.e.) as follows[58]:

$$b = z/d \tag{2}$$

$$\text{s.e.} = 1/d \tag{3}$$

where

$$d = \sqrt{2f(1-f)(z^2 + N)} \tag{4}$$

and *N* is the sample size.

### Prioritizing probable mediating genes at *trans*-pQTLs

To prioritize probable mediating genes at *trans*-pQTLs, we used the ProGeM tool[18]. To identify cis-eQTLs that could mediate *trans*-pQTLs, we queried the *trans*-pQTL sentinel variants in eQTLGen[14], the eQTL Catalogue[17] and the GTEx (v.8) data. To determine whether the *trans*-pQTL sentinel variants are likely to be causal cis-eQTL variants in the eQTL Catalogue and GTEx data, we used the fine-mapped eQTL credible sets available at the eQTL Catalogue (https://www.ebi.ac.uk/eqtl/Data_access). For the eQTLGen data, where credible sets were not available, we used a manual approach whereby we: (1) first defined a region around each *trans*-pQTL sentinel variant of ±500 kb; (2) identified the variant with the lowest cis-eQTL *P* value in this region for the cis-affected gene(s); and (3) checked to see whether this sentinel cis-eQTL variant was the same sentinel variant for the *trans*-pQTL, or if the two were in high LD ($r^2 \geq 0.8$).

For the 'top-down' component of ProGeM, we first identified locally encoded genes using a window around each *trans*-pQTL sentinel variant of ±500 kb. We then probed the proteins encoded by these local genes using: (1) protein–protein interaction (PPI) data and (2) data from functional annotation databases. With the PPI data, we sought to determine whether there was evidence to indicate that genes residing close to each sentinel variant might interact with the corresponding *trans*-affected protein. We used the Bioconductor package STRINGdb (v.2.8.4) to identify any pairwise interactions. We used data from functional annotation databases to determine whether any local genes encode proteins that might be functionally related to the corresponding *trans*-affected protein(s). For both the *trans*-affected proteins and the locally encoded proteins, all assigned GO terms, Reactome pathways and KEGG (Kyoto Encyclopedia of Genes and Genomes) pathways were extracted using the Bioconductor biomaRt (v.2.52) and KEGGREST (v.1.36) packages. To assess whether there was significant overlap between the functional annotation terms/pathways assigned to locally encoded proteins and the corresponding *trans*-affected proteins, we determined the number of shared and nonshared terms for each local gene and the corresponding *trans*-affected protein. Fisher's exact test was then applied for each local gene–*trans*-protein combination and *P* values were Bonferroni corrected for the number of local genes at each given *trans*-pQTL. The background set of terms for each *trans*-pQTL was composed of all terms assigned to all local genes at the locus (that is, all protein-coding genes within 500 kb of the sentinel variant).

To determine the most likely mediating genes for the multi-locus-regulated proteins IL-12B, KITLG and TNFSF10 (TRAIL), we used the STRINGdb webtool to identify interactions or functional relationships between genes residing at distinct loci. This is based on the concept that, if the mediating genes at distinct loci are all associated with plasma levels of the same protein, then they may share some other functional relationship. As input to STRINGdb, we used all proteins encoded by candidate mediating genes identified by ProGeM (Supplementary Table 9) at each of the loci for a given protein, as well as the relevant *trans*-affected protein. We deemed clusters of proteins residing at distinct loci with multiple functional interactions to be the most likely mediating genes at their respective loci. We performed a phenome-scan of the *trans*-pQTLs for *KITLG* using the Open Targets Genetics webtool[59].

### Overlap of pQTL and disease traits

We used a PhenoScanner v.2-based R code to look up associations of our pQTL sentinels and their LD proxies ($r^2 \geq 0.8$) in disease GWAS summary statistics.

To investigate potential colocalization between a *trans*-pQTL (rs2364485) for LTA identified in our meta-analysis, a multiple sclerosis GWAS signal[25] and a cis-eQTL for *LTBR* from eQTLGen[14], we used HyPrColoc for multi-trait colocalization[60]. We obtained multiple

sclerosis summary statistics (MSchip, 'discovery_metav3.0.meta.gz') from Patsopoulos et al.[25] by a request to the International Multiple Sclerosis Genetics Consortium. Due to a lack of genotype coverage at the *LTBR/TNFRSF1A* locus in the extended and replication samples from Patsopoulos et al., we selected the summary statistics from the 'discovery' sample ($n = 41,505$) for colocalization analyses, not the full meta-analysis. As a result, the *P* value for association between the variant of interest (rs2364458) and multiple sclerosis in the discovery subset ($P = 5.78 \times 10^{-6}$, logistic regression) was higher than reported in Patsopoulos et al.[25] ($P = 2.0 \times 10^{-20}$, fixed-effects meta-analysis). We then extracted summary statistics for rs2364458 (±1 Mb) (chr12: 5514963-7514963) from each of the three datasets and performed conditional analyses to adjust for any independent signals at the locus using GCTA-COJO. We ran this using a two-step approach: we first used the COJO-slct function to identify independent signals at the locus and then, for datasets with signals independent of rs2364485, we used COJO-cond to generate conditioned summary statistics for use in HyPrColoc. HyPrColoc returns the posterior probability that two or more traits colocalize, akin to PP4 from coloc. We considered a PP ≥ 0.8 as robust evidence of colocalization between traits.

## MR analyses

We performed MR analyses using the proteins with *cis*-pQTLs identified in this meta-analysis as exposures and IMDs as outcomes. All MR analyses were run using the GSMR method[28], which implements two-sample MR accounting for correlation between variants. For each protein analyzed, we defined a ±1-Mb window around the gene encoding it and extracted pQTL summary statistics for this region. For outcome data, we downloaded GWAS summary statistics for IMDs from OpenGWAS (https://gwas.mrcieu.ac.uk/datasets) or the GWAS catalog (https://www.ebi.ac.uk/gwas/downloads), where studies with larger sample sizes or more variants were available. For IMDs with several alternative datasets available, we selected the dataset with the largest number of cases, provided that it: (1) had genotype data with sufficient coverage at the loci of interest, (2) was generated in European-ancestry samples so that it matched the ancestry of the participants in our pQTL meta-analysis and (3) had effect estimates and s.e. values either available or calculable. Proteins encoded by genes in the *HLA* region were excluded because MR analysis would be confounded by complex LD. The analysis involved 57 proteins and 14 diseases. We used the GSMR implementation in GCTA with the following parameters: (1) at least three (--gsmr-snp-min 3) genome-wide significant (--gwas-thresh 5e-8), quasi-independent variants (--clump-r2 0.1); (2) difference in the allele frequency of each effect allele between the GWAS summary datasets and the LD reference sample of at most 0.4 (--diff-freq 0.4); and (3) a *P*-value threshold of 0.05 for the HEIDI-outlier filtering analysis (--heidi-thresh 0.05), which is used to identify potential confounding by LD (https://yanglab.westlake.edu.cn/software/gcta/#Mendelianrandomisation). The *P* values were corrected for the number of models tested using the Benjamini–Hochberg correction, with FDR < 0.01 used to define statistical significance.

To evaluate the robustness of significant associations, we performed additional checks. First, we checked the strength of the disease association in the GWAS summary statistics. Of the 22 significant, protein–disease MR associations, we eliminated 5 due to the lack of convincing disease association (smallest *P* value at the locus >$1 \times 10^{-4}$). For the remaining 17 MR associations, we then evaluated whether there might be confounding due to LD. We first evaluated $r^2$ between the sentinel pQTL and the disease-associated variant. For 12 of 17 disease–protein pairs, $r^2$ was >0.8 (Supplementary Table 14). We next performed visual inspection of regional association plots of these 12 pQTL–disease pairs (Supplementary Fig. 4) and colocalization testing using pairwise conditional and colocalization analysis (PWCoCo)[61,62], which accounts for the presence of multiple independent signals within a locus (see below).

## PWCoCo

PWCoCo[61,62] integrates conditional analyses (GCTA-COJO) to identify independent signals for each of two tested traits associated with a genomic region, followed by pairwise colocalization analyses (COLOC) to test all possible pairs of conditionally independent signals across the traits. We ran PWCoCo for the 12 significant protein–disease pairs that resulted from our MR-filtering steps using the default parameters, detailed as follows: (1) *P*-value cutoff for variants to be selected by the stepwise selection process, --p_cutoff $5 \times 10^{-8}$ for disease and protein summary statistics; (2) a large number of variants subject to the stepwise selection process, --top_snp $1 \times 10^{-10}$; (3) distance beyond which variants are treated as in linkage equilibrium, --ld_window $1 \times 10^{-7}$ (kb); (4) collinearity threshold for variants, --collinear 0.9; (5) variant frequency filter for the reference dataset according to this threshold, --maf 0.1; (6) exclusion threshold for variants with allele frequency difference between the phenotype and the reference datasets, --freq_threshold 0.2; (7) stop criteria, --init_h4 80 (that is. 80%); and (8) the three prior probabilities, --coloc_pp $1 \times 10^{-4}$, $1 \times 10^{-4}$ and $1 \times 10^{-5}$.

## *CXCL5* differential expression analysis in UC cohorts

Changes in *CXCL5* gene expression levels were evaluated in four independent cohorts, including the IBD TaMMA platform[29], the GEO series, accession nos. GSE16879 and GSE206285, and the Imperial UC cohort. IBD TaMMA (https://ibd-meta-analysis.herokuapp.com) gives access to 3,853 transcriptomic profiles from 26 independent studies including IBD and control samples across different tissues, all processed with the same pipeline and batch corrected[29]. Pre-computed differential expression results between colon biopsies from patients with UC versus healthy donors were downloaded and plotted.

Data from Gene Expression Omnibus (GEO) accession no. GSE16879 used in the present study consist of colonic mucosa microarray expression profiles from healthy donors ($n = 6$) and patients with UC ($n = 24$) sampled before the first infliximab treatment[63]. CEL file import into R, and background correction, RMA (Robust Multiarray Averaging) normalization of the raw intensity data were carried out using the oligo package. Only probe sets with median expression >4 and uniquely associated with a single ENTREZ gene identifier were kept for analysis. Intensity data for different probe sets mapped to the same ENTREZ gene identifier were combined by taking the geometric mean sample wise. Tests of differential gene expression of UC samples compared with healthy control samples were performed using the limma package. *P* values were adjusted for multiple testing using the Benjamini–Hochberg procedure.

GEO accession no. GSE206285 contains array-based transcriptomic data collected at baseline as part of UNIFI, a randomized, placebo-controlled, phase 3 clinical trial evaluating the efficacy and safety of ustekinumab[64]. RMA signal intensity profiles and associated donor information were downloaded from NCBI's GEO. Only probe sets associated to only one ENTREZ gene identifier were kept for analysis. Intensity data for different probe sets mapped to the same ENTREZ gene identifier were combined by taking the geometric mean sample wise. Genes with median expression >3 across all samples were tested for differential expression between UC samples ($n = 550$) versus healthy control samples ($n = 18$) using the R limma package. P values were adjusted for multiple testing with the Benjamini–Hochberg procedure.

The Imperial UC cohort includes whole-tissue biopsies from patients with UC ($n = 16$) and healthy volunteers ($n = 6$). RNA was extracted (QIAGEN RNeasy mini-kit) and sequencing libraries were generated using NEBNext Ultra RNA Library Prep Kit for Illumina (New England Biolabs (NEB)) following the manufacturer's recommendations. Briefly, mRNA was purified from total RNA using poly(T) oligo-attached magnetic beads. Fragmentation was carried out using divalent cations under an elevated temperature in NEBNext First Strand Synthesis Reaction Buffer (5×). First-strand complementary DNA was synthesized using random hexamer primer and M-MuLV reverse transcriptase

(RNase H). Second-strand cDNA synthesis was subsequently performed using DNA polymerase I and RNase H. Remaining overhangs were converted into blunt ends via exonuclease/polymerase activities. After adenylation of 3′-ends of DNA fragments, the NEBNext Adapter with hairpin loop structure was ligated to prepare for hybridization. Library fragments were purified with AMPure XP system (Beckman Coulter) and treated with 3 µl of USER Enzyme (NEB) at 37 °C for 15 min, followed by 5 min at 95 °C. Then PCR was performed with Phusion High-Fidelity DNA polymerase, universal PCR primers and index (X) primer. Library quality was assessed on Agilent Bioanalyzer 2100 and Nanodrop ND-1000 Spectrophotometer. The library preparations were sequenced on an Illumina HiSeq platform, generating 150-bp paired-end reads. The resulting fastq files were processed with trim-momatic[65] (v.0.39) to remove adapter contamination and poor-quality bases. The output read files were mapped to the GRCh38 assembly of the human genome using Hisat2 (ref. [66]) (v.2.2.1) with default parameters. The number of reads mapping to the genomic features annotated in Ensembl with a MAPQ score ≥10 was calculated for all samples using htseq-count[67] (v.0.11.3) with default parameters. Data for Ensembl genes with no associated ENTREZ gene identifier were discarded; the read counts for Ensembl genes mapped to the same ENTREZ gene identifier were summed up sample wise. Differential expression analysis between UC versus healthy biopsies was performed in R (v.3.6.1) using Wald's test as implemented in DESeq2. Only genes with an average read count across samples ≥10 were tested for differential expression. P values were adjusted for multiple testing using the Benjamini–Hochberg procedure.

## Reporting summary

Further information on research design is available in the Nature Portfolio Reporting Summary linked to this article.

## Data availability

Full per-protein GWAS summary statistics are available for download at https://www.phpc.cam.ac.uk/ceu/proteins and the EBI GWAS Catalog (accession numbers GCST90274758 to GCST90274848). Individual-level genetic and proteomic data available for the INTERVAL cohort are deposited in the European-Genome Phenome Archive under accession no. EGAS00001002555. Gene expression data are in GEO under accession no. GSE16879 for mucosal expression in patients with IBD (https://www.ncbi.nlm.nih.gov/geo/query/acc.cgi?acc=GSE16879) and GSE206285 for the UNIFI trial (https://www.ncbi.nlm.nih.gov/geo/query/acc.cgi?acc=GSE206285) and in the the IBD TaMMA (https://ibd-meta-analysis.herokuapp.com). Whole-blood cis-eQTL summary statistics from the eQTLGen Consortium were downloaded from https://www.eqtlgen.org/cis-eqtls.html. Fine-mapped eQTL credible sets were downloaded from the eQTL Catalogue (https://www.ebi.ac.uk/eqtl/Data_access). MR GWAS summary statistics for IMDs were downloaded from OpenGWAS (https://gwas.mrcieu.ac.uk/datasets) or the GWAS catalog (https://www.ebi.ac.uk/gwas/downloads).

## Code availability

GitHub: https://jinghuazhao.github.io/INF; cambridge-ceu: https://cambridge-ceu.github.io/public (modified METAL, pQTLtools).

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

## Acknowledgements

This work was performed under the auspices of the SCALLOP Consortium. We thank the following: study participants from the contributing cohorts; the International Multiple Sclerosis Genetics Consortium, which provided multiple sclerosis GWAS summary statistics used in our analyses; A. Siopi and D. McLeod for support with SCALLOP Consortium administration; the authors of the GCTA software for advice; B. Prins for help with the INTERVAL study genotype data quality control; and A. Richard for comments on the manuscript. J.E.P was supported by a grant and fellowship from the Medical Research Foundation (grant nos. MRF-042-0001-RG-PETE-C0839 and MRF-057-0003-RG-PETE-C0799). E.J.N. was supported by the Schmidt Science Fellows, in partnership with the Rhodes Trust. P.S. was supported by a Rutherford Fund Fellowship from the UK Medical Research Council (MRC; grant no. MR/S003746/1). J.D. holds a British Heart Foundation Professorship and a National Institute for Health and Care Research (NIHR) Senior Investigator Award*. C. Ha is supported by an MRC University Unit Programme Grant 'QTL in Health and Disease' (grant no. U.MC_UU_00007/10). Funding of the GWASs and proteomics studies of STABILITY and ARISTOTLE were supported by GlaxoSmithKline, BristolMyersSquibb and the Swedish Foundation for Strategic Research (grant no. RB13-0197). The Orkney Complex Disease Study (ORCADES) was supported by the Chief Scientist Office of the Scottish Government (grant nos. CZB/4/276 and CZB/4/710), a Royal Society University Research Fellowship to J.F.W., the MRC Human Genetics Unit quinquennial program 'QTL in Health and Disease',

Arthritis Research UK and the European Union framework program 6 EUROSPAN project (contract no. LSHG-CT-2006-018947). DNA extractions were performed at the Edinburgh Clinical Research Facility, University of Edinburgh. We acknowledge the invaluable contributions of the research nurses in Orkney, the administrative team in Edinburgh and the people of Orkney. For the purpose of open access, the author has applied a Creative Commons Attribution (CC BY) license to any author-accepted manuscript version arising from this submission. Participants in the INTERVAL trial were recruited with the active collaboration of National Health Service (NHS) Blood and Transplant England (www.nhsbt.nhs.uk), which has supported field work and other elements of the trial. DNA extraction and genotyping were co-funded by the NIHR, the NIHR BioResource (http://bioresource.nihr.ac.uk) and the NIHR Cambridge Biomedical Research Centre (grant no. BRC-1215-20014). The academic coordinating center for INTERVAL was supported by core funding from: the NIHR Blood and Transplant Research Unit (BTRU) in Donor Health and Genomics (grant no. NIHR BTRU-2014-10024), NIHR BTRU in Donor Health and Behaviour (grant no. NIHR203337), MRC (grant no. MR/L003120/1), British Heart Foundation (grant nos. SP/09/002, RG/13/13/30194 and RG/18/13/33946) and NIHR Cambridge BRC (grant nos. BRC-1215-20014 and NIHR203312)* and has received funding from a European Commission Innovative Medicines Initiative (BigData@Heart). The academic coordinating center thank blood donor center staff and blood donors for participating in the INTERVAL trial. This work was supported by Health Data Research UK, which is funded by the MRC, Engineering and Physical Sciences Research Council, Economic and Social Research Council, Department of Health and Social Care (England), Chief Scientist Office of the Scottish Government Health and Social Care Directorates, Health and Social Care Research and Development Division (Welsh Government), Public Health Agency (Northern Ireland), British Heart Foundation and Wellcome. Estonian Biobank work was supported by the European Regional Development Fund and the program Mobilitas Pluss (MOBTP108, grant nos. 2014-2020.4.01.15-0012 GENTRANSMED and 2014-2020.4.01.16-0125). The present study was also funded by the EU H2020 (grant no. 692145), the Estonian Research Council (grant nos. PUT1660 and PUT PRG1291). Data analyses with Estonian datasets were carried out in part in the High-Performance Computing Center of the University of Tartu. The SWEBIC biobank was supported by the Stanley Medical Research Institute. The proteomic analyses in SWEBIC were funded by the Swedish foundation for Strategic Research (grant no. KF10-0039). For RECOMBINE and SWEBIC, the data handling and analysis were enabled by resources provided by the Swedish National Infrastructure for Computing (SNIC), partially funded by the Swedish Research Council through grant no. 2018-05973. The CROATIA-Vis study was funded by grants from the UK MRC, the Republic of Croatia Ministry of Science, Education and Sports (grant nos. 108-1080315-0302 and 216-1080315-0302) and the Croatian Science Foundation (grant no. 8875). We thank the staff of several institutions in Croatia who supported the field work, including Zagreb Medical Schools, the Institute for Anthropological Research in Zagreb, the recruitment team from the Croatian Centre for Global Health, University of Split and all the study participants. The KORA study was initiated and financed by the Helmholtz Zentrum München—German Research Center for Environmental Health, which is funded by the German Federal Ministry of Education and Research and the State of Bavaria. Furthermore, KORA research was supported within the Munich Center of Health Sciences, Ludwig-Maximilians-Universität, as part of LMUinnovativ. The measurement of inflammatory biomarkers was funded by a grant from the German Center for Diabetes Research (DZD; to C. Herder and B. Thorand). This work was also supported by the Ministry of Culture and Science of the State of North Rhine-Westphalia and the German Federal Ministry of Health. The present study was supported in part by a grant from the German Federal Ministry of Education and Research to the DZD. N.P. is supported by a Wellcome Trust Discovery award (no. 225875/Z/22/Z). D.C. is supported by the NIHR Imperial Biomedical Research Centre (BRC)*. Infrastructure support for this research was provided by the NIHR Imperial BRC. Support for title page creation and format was provided by AuthorArranger, a tool developed at the National Cancer Institute. We acknowledge the Danish node of the TRYGGVE server and the University of Cambridge's High Performance Computing cluster, on which computations were performed. *The views expressed are those of the author(s) and not necessarily those of the NHS, the NIHR, NHSBT or the Department of Health and Social Care.

## Author contributions

J.E.P., N.E., E.M.-D., A.K.H., A.K., S.E., L.F., C. Herder, L.J., S.E.B. and P.S. conducted study-level analyses. C.G., D.S.P., O.P., B.T., H.G., M.R., U.V., T.O., C. Hayward, A.J., U.G., N.P., O.H., N.M.-C., P.K.J., J.D., L.P., L.K., M.L., J.F.W., A.S., L.W., A.M., A.S.B. and J.E.P. provided data and study supervision. D.C., J.D.-B. and N.P. collected IBD samples and generated and analyzed the IBD RNA-sequencing data. J.H.Z., D.S., A.K., J.M. and P.S. conducted the meta-analysis and downstream analyses. J.H.Z., D.S., E.N., A.S.B. and J.E.P. drafted the manuscript. J.F.W., A.M., A.S.B. and J.E.P. conceived the project. All authors critically reviewed the manuscript and gave final approval to publish.

## Competing interests

J.D. serves on scientific advisory boards for AstraZeneca, Novartis and UK Biobank, and has received multiple grants from academic, charitable and industry sources outside of the submitted work. A.S.B. has received grants unrelated to this work from AstraZeneca, Bayer, Biogen, BioMarin, Bioverativ, Novartis and Sanofi. J.E.P. has received hospitality and travel expenses to speak at Olink-sponsored academic meetings (none within the past 5 years). During the drafting of the manuscript, D.S.P. became a full-time employee of AstraZeneca and P.S. became a full-time employee of GlaxoSmithKline. M.L. has received lecture honoraria from Lundbeck pharmaceutical. The other authors declare no competing interests.

## Additional information

**Extended data** is available for this paper at https://doi.org/10.1038/s41590-023-01588-w.

**Correspondence and requests for materials** should be addressed to Adam S. Butterworth or James E. Peters.

# Analysis outline

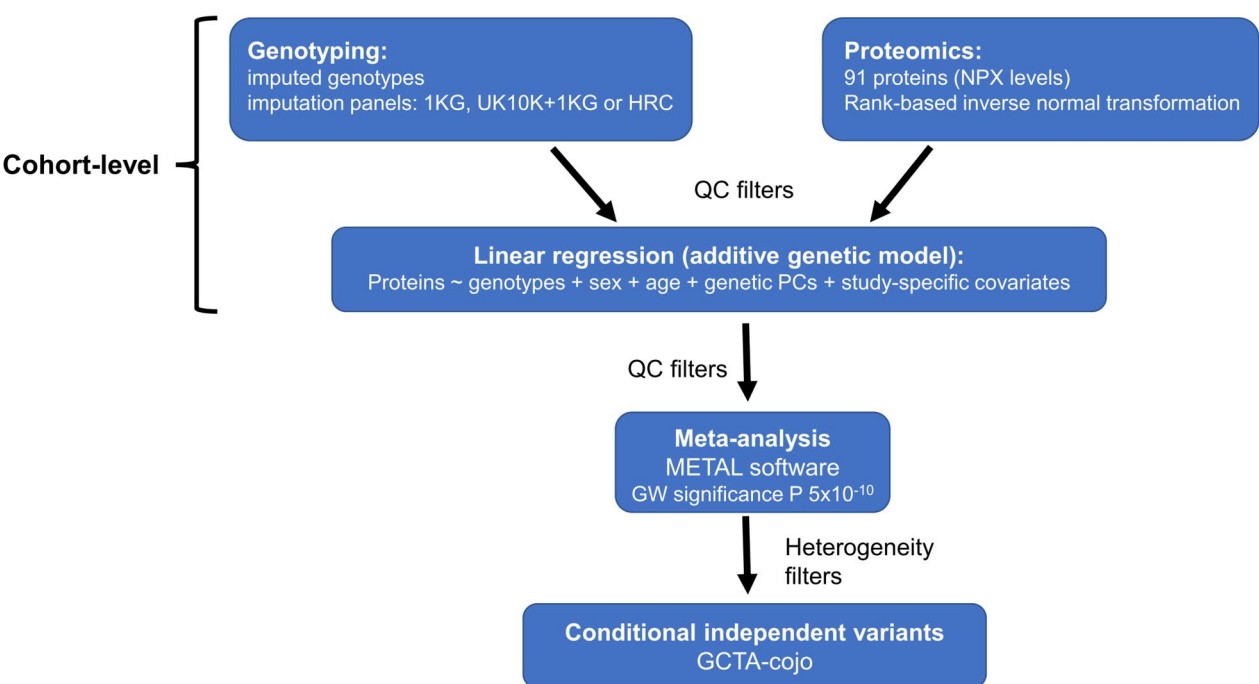

**Extended Data Fig. 1 | Overview of the pQTL analysis.** Schematic of the analysis pipeline.

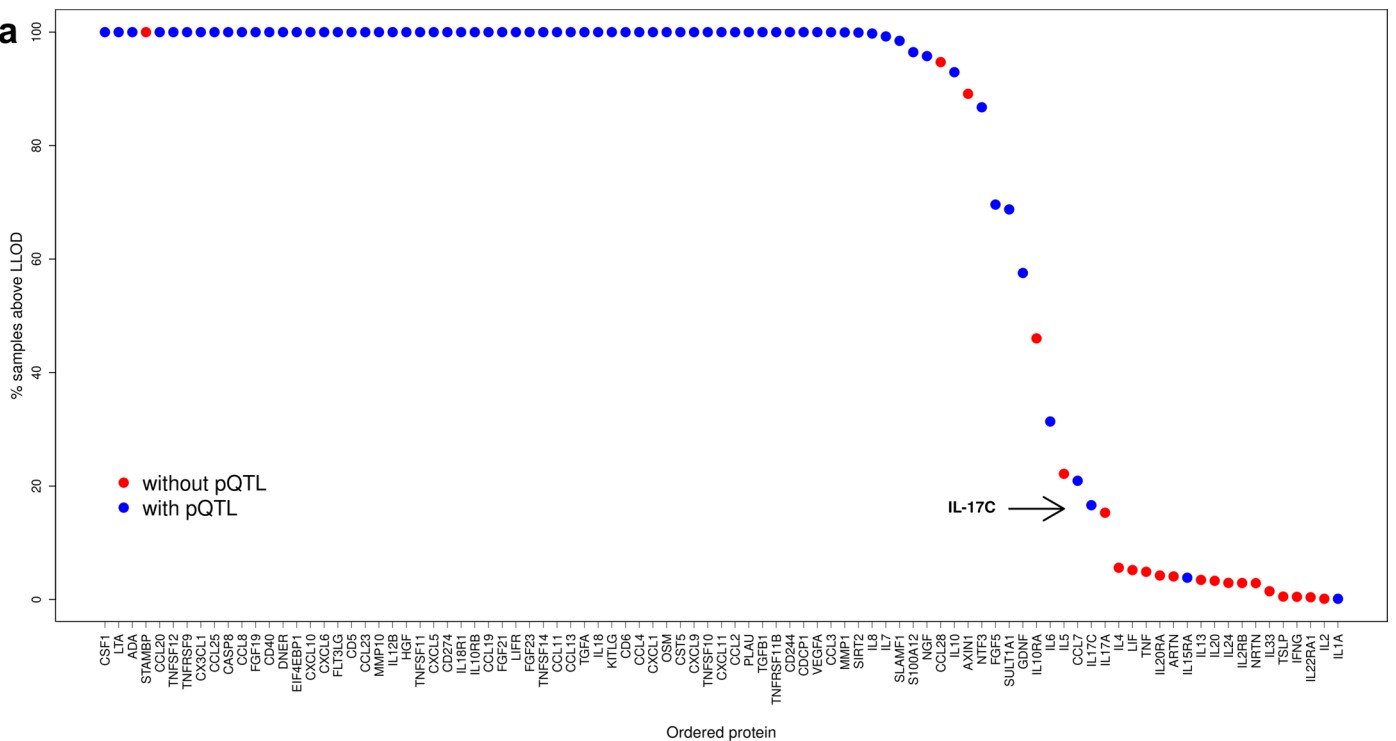

**Extended Data Fig. 2 | Plasma protein abundance and pQTL detection.**
**a)** Proteins with low abundance are more likely to have no detectable pQTL.
Y-axis: percentage of samples above lower limit of detection for each protein,
calculated using the INTERVAL data (n = 4,896) for which we had individual-level
protein data available. Blue and red points indicate presence or absence of at
least 1 significant pQTL in the GWAS meta-analysis, respectively. **b)** Manhattan
plot for genetic associations with plasma IL17C, where the red horizontal line
indicates the statistical significance threshold ($5 \times 10^{-10}$). P-values from linear
regression.

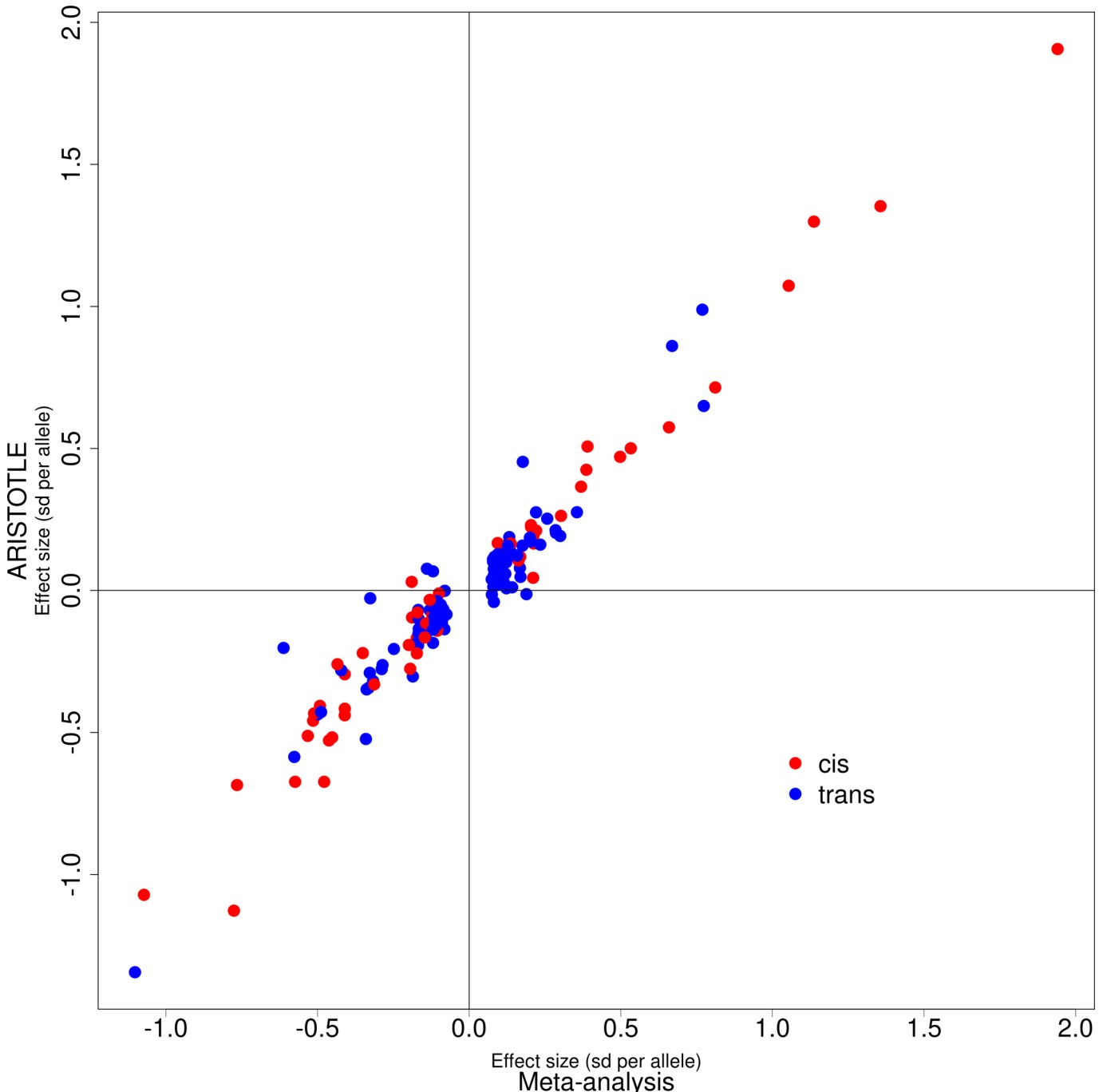

**Extended Data Fig. 3 | pQTL replication in the ARISTOTLE cohort.** Comparison of effect sizes between pQTLs from the discovery pQTL meta-analysis (n = 14,824) and the ARISTOTLE cohort (n = 1,585). Each point represents a genetic variant that was a significant pQTL in the discovery meta-analysis. Effect size = standard deviation (sd) increase in protein per allele. 174 of 180 genetic variants were available for testing in the ARISTOTLE data. Red= cis, Blue= trans.

**a**

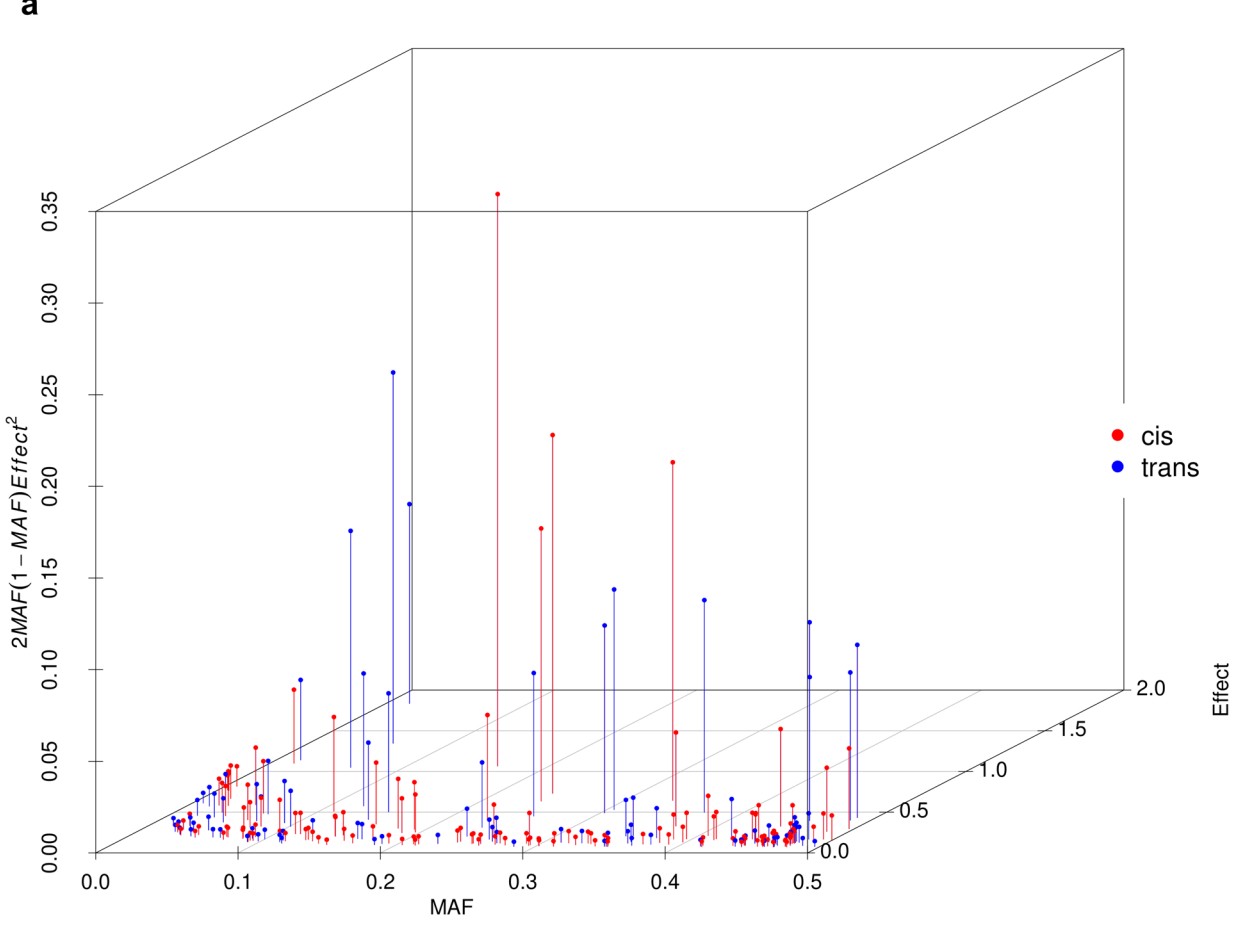

**b**

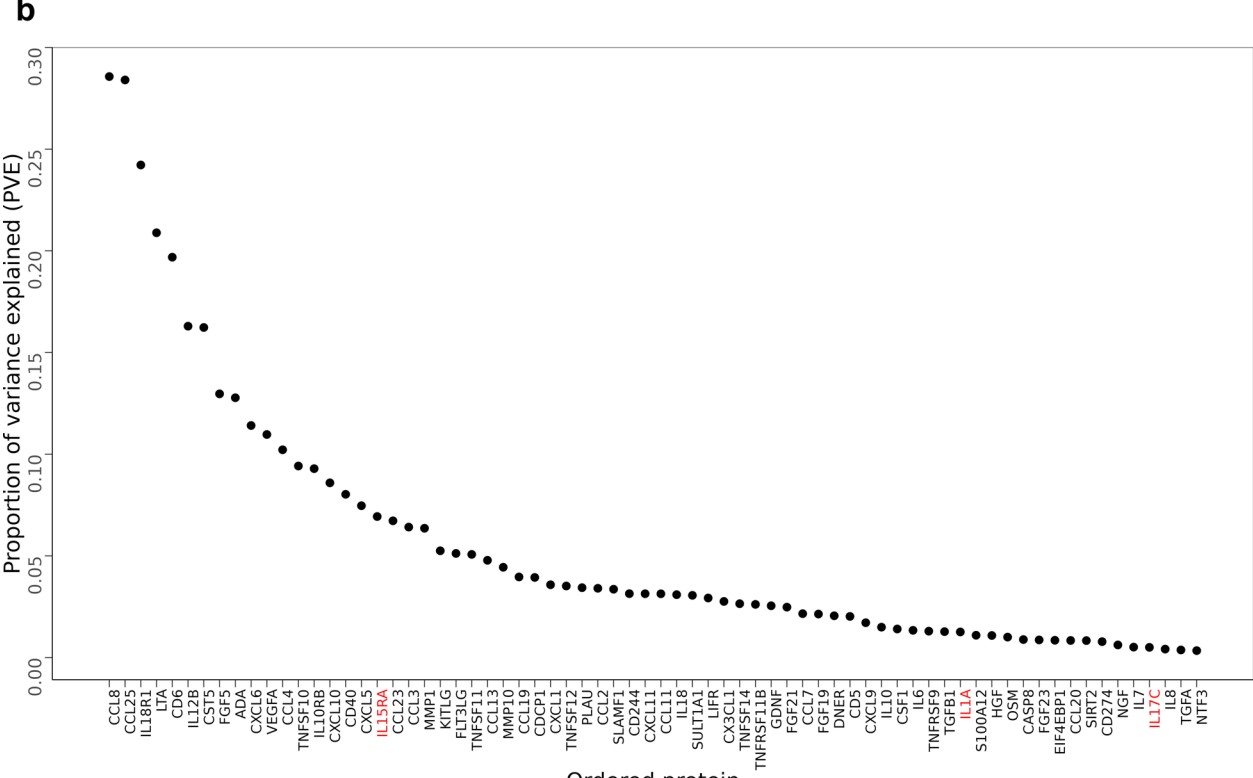

**Extended Data Fig. 4 | Genetic architecture of circulating inflammation-related proteins. a**) Relationship between minor allele frequency (MAF), pQTL effect size and proportion of variance explained (2MAF(1-MAF)Effect$^2$), for 227 conditionally independent pQTLs (red=cis, blue=trans). **b**) Proportion of variance explained (PVE) by the conditionally independent variants associated with each protein. Proteins are annotated using the gene symbol of their encoding genes. Protein names are coloured in red if over 80% of samples have levels below the lower limit of detection in the INTERVAL dataset.

| Outcome | b | SE | | 95%CI | P |
|---|---|---|---|---|---|
| CCL2 | 0.169 | 0.011 | | [ 0.147; 0.192] | 8.39e-51 |
| CCL7 | -0.090 | 0.013 | | [-0.115; -0.064] | 4.67e-12 |
| CCL8 | -0.097 | 0.012 | | [-0.120; -0.075] | 4.95e-17 |
| CCL11 | 0.075 | 0.011 | | [ 0.053; 0.097] | 5.03e-11 |
| CCL13 | 0.189 | 0.011 | | [ 0.167; 0.211] | 9.89e-62 |
| CXCL6 | 0.082 | 0.011 | | [ 0.059; 0.104] | 1.29e-12 |
| WBC | 0.016 | 0.002 | | [ 0.012; 0.019] | 5.52e-17 |
| Monocyte count | 0.027 | 0.002 | | [ 0.023; 0.031] | 1.40e-46 |
| Basophil count | 0.028 | 0.002 | | [ 0.024; 0.032] | 5.32e-41 |

-0.2  -0.1  0  0.1  0.2

Effect size

**Extended Data Fig. 5 | Chemokine *trans*-pQTL hotspot.** Forest plot showing the associations for the pleiotropic *trans*-pQTL at rs12075 (GRCh37, 1:158175353-160525679) with plasma levels of chemokines and blood cell counts. Center of bar = effect size estimate, whiskers = 95% confidence interval (cI). WBC = white blood cell count. P = p-value, b= beta (effect size). SE = standard error. Blood cell association data from Chen et al.[20]. P-values from linear regression.

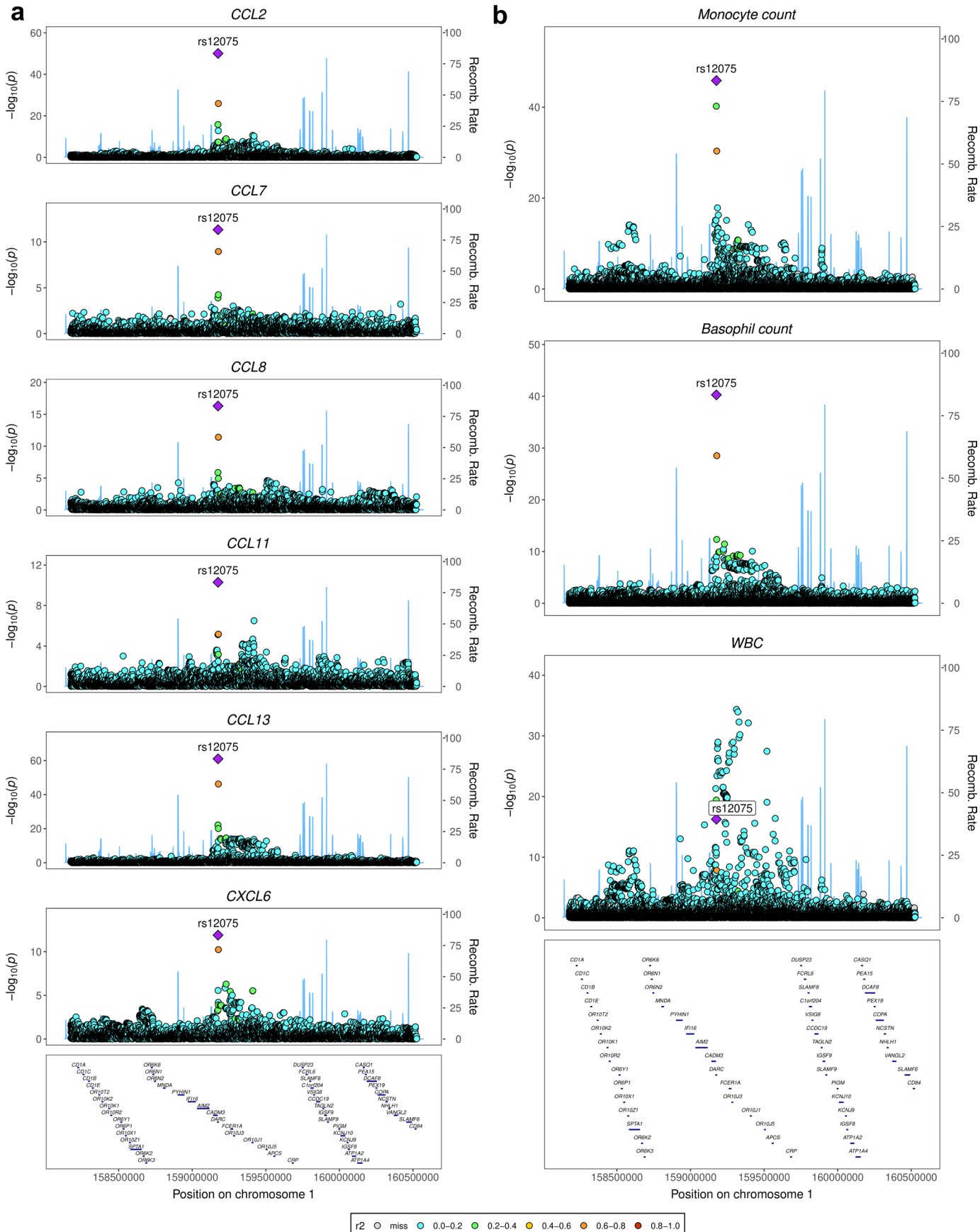

**Extended Data Fig. 6 | Colocalisation of pleiotropic chemokine trans-pQTL and blood cell count trait signals.** Regional association plots in the region around rs12075 (GRCh37, 1:158175353-160525679). **a**, Association with plasma chemokine levels. **b**, Associations with basophil, monocyte and white blood cell (WBC) counts using data from Chen *et al*.[20]. P-values from linear regression.

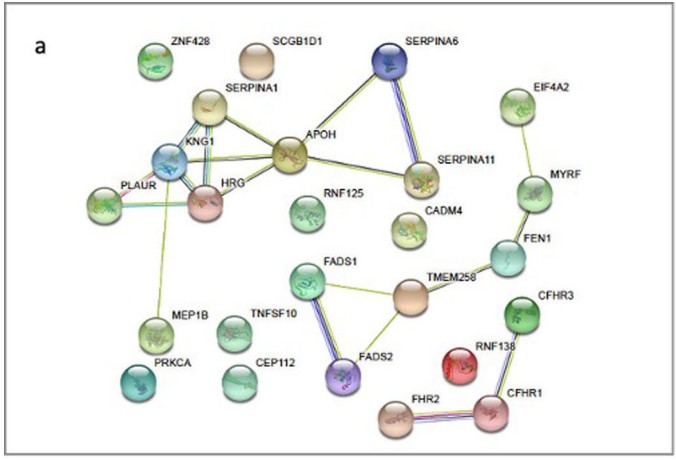

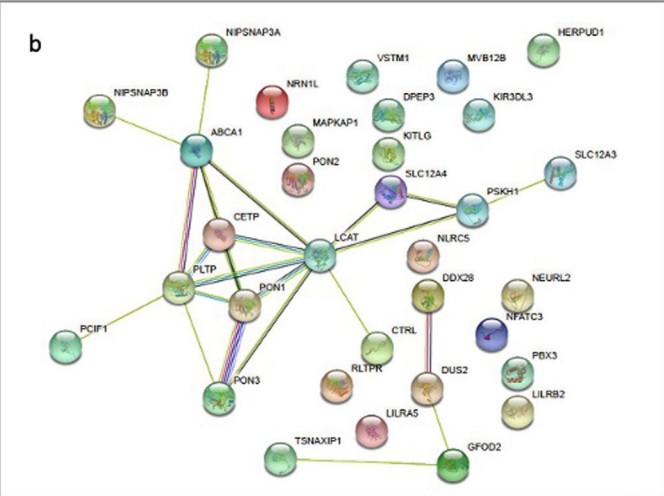

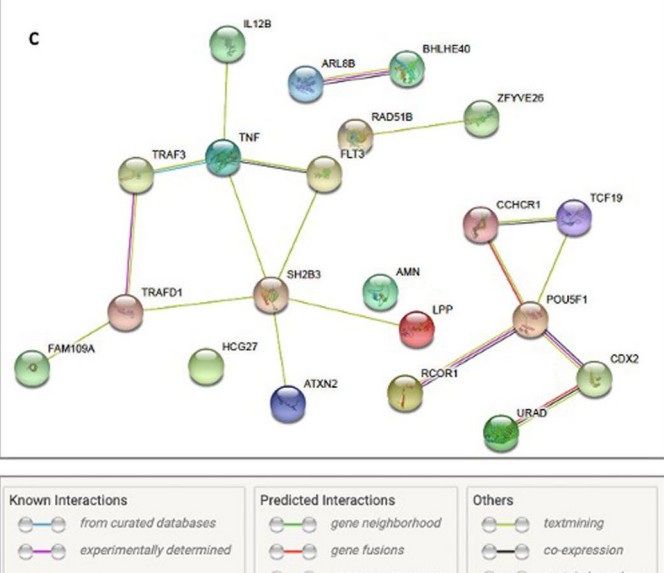

**Extended Data Fig. 7 | Interactions between the candidate mediators for multi-locus-regulated proteins. a**) TNFSF10 (also known as TRAIL), **b**) KITLG (also known as stem cell factor), and **c**) IL12B. The graphs were generated using the STRINGdb (v11.5) webtool. The colouring of the edges indicates the type of evidence supporting an interaction, as shown in the legend above.

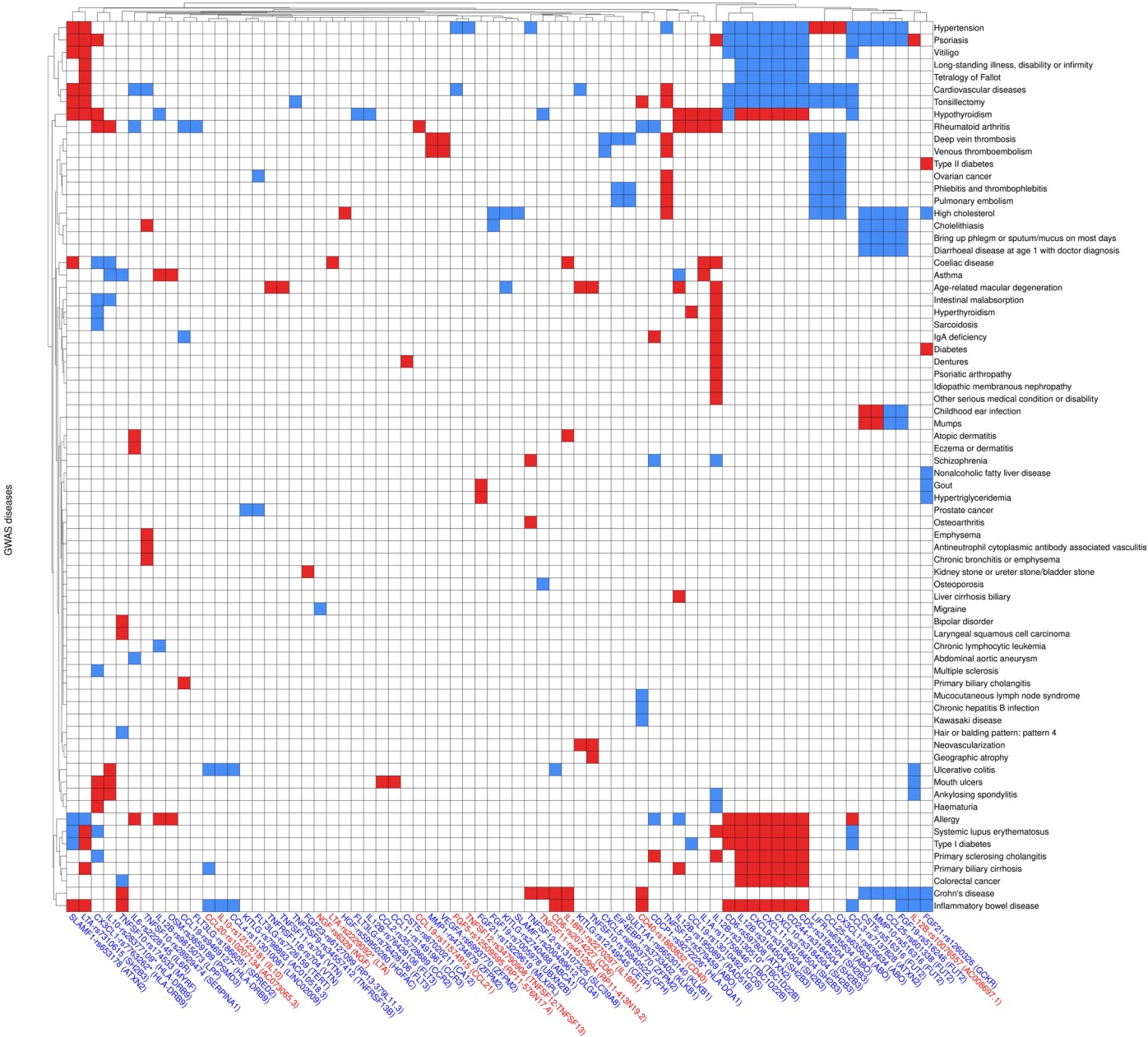

**Extended Data Fig. 8 | Protein-disease connections from overlap of pQTLs and disease GWASs.** The protein and the corresponding pQTL sentinel variant are indicated in the format of protein-rsid. The nearest gene to the pQTL sentinel variant is shown in brackets. Red lettering= cis-pQTL, blue lettering= trans-pQTL. Asterix indicates the genetic variant lies in the *HLA* region. Red squares: genetic susceptibility to increased plasma levels of the protein is associated with increased disease risk. Blue squares: decreased disease risk.

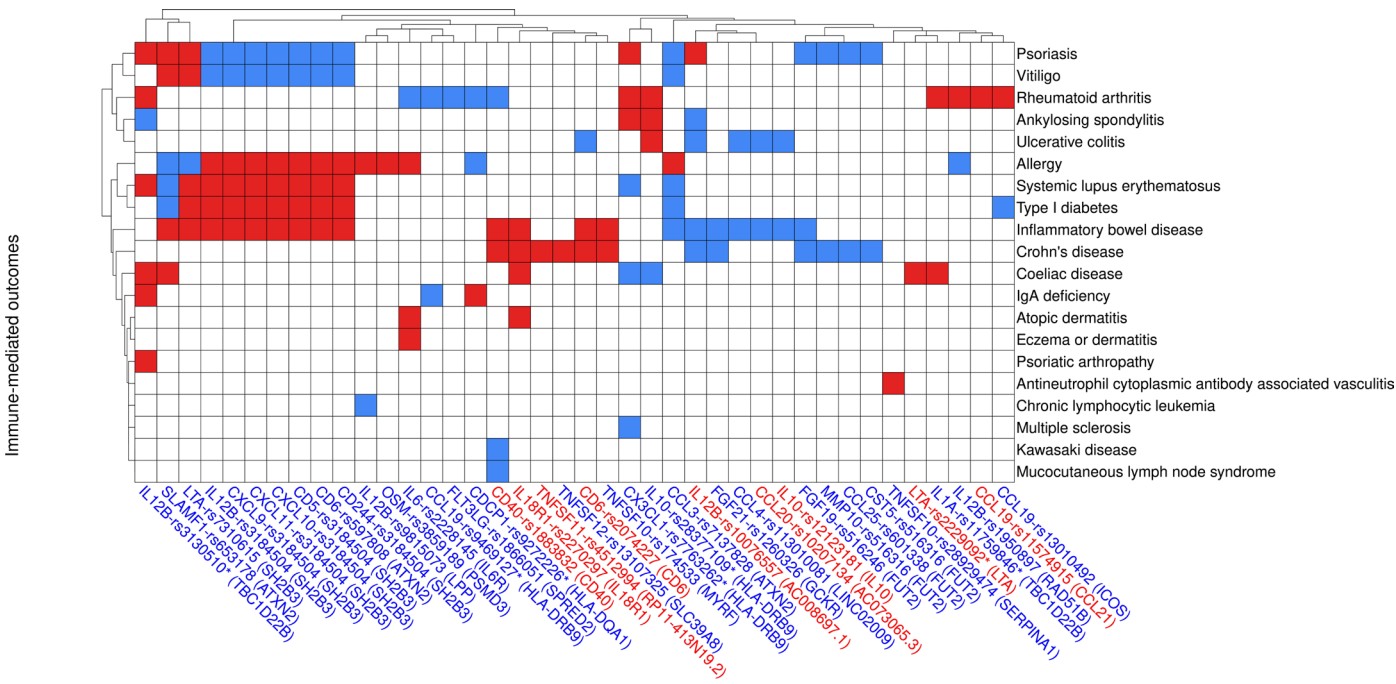

Protein-pQTL (Nearest gene)

**Extended Data Fig. 9 | Protein and immune-mediated disease (IMD) connections from overlap of pQTLs and disease GWASs.** The protein and the corresponding pQTL sentinel variant are indicated in the format of protein-rsid. The nearest gene to the pQTL sentinel variant is shown in brackets. Red lettering= cis-pQTL, blue lettering= trans-pQTL. Asterix indicates the genetic variant lies in the *HLA* region. Red squares: genetic susceptibility to increased plasma levels of the protein is associated with increased disease risk. Blue squares: decreased disease risk.

**a**

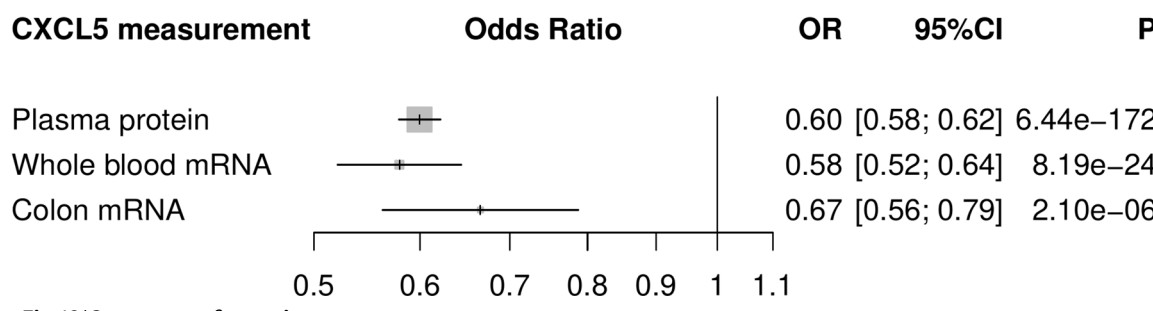

**b**

| CXCL5 measurement | Odds Ratio | OR | 95%CI | P |
|---|---|---|---|---|
| Plasma protein | | 0.60 | [0.58; 0.62] | 6.44e−172 |
| Whole blood mRNA | | 0.58 | [0.52; 0.64] | 8.19e−24 |
| Colon mRNA | | 0.67 | [0.56; 0.79] | 2.10e−06 |

**Extended Data Fig. 10 | See next page for caption.**

**Extended Data Fig. 10 | Mendelian randomisation analysis for CXCL5 and ulcerative colitis. a**) Scatterplot showing the 13 variants used in the GSMR analysis assessing the effect of CXCL5 on ulcerative colitis (UC) risk from the GWAS by de Lange *et al* (ref. 51) Each point represents a genetic variant, and indicates the effect size of the variant on CXCL5 levels versus UC risk (log odds ratio). Vertical and horizontal lines represent 95% confidence intervals. **b**) Directional concordance between CXCL5 pQTL and blood and colon tissue eQTLs. Forest plots showing effect size estimates for rs450373 pQTL in plasma (from our discovery meta-analysis) and eQTLs in whole blood and transverse colon tissue (GTEx v8 data). OR= odds ratio, calculated from beta estimate (representing the change in inverse-rank normalised plasma protein level in standard deviations associated with each copy of the effect allele). CI = confidence interval. P = p-value. Centre of bar = OR estimate, whiskers = 95% CI.

# Reporting Summary

## Statistics

For all statistical analyses, confirm that the following items are present in the figure legend, table legend, main text, or Methods section.

| n/a | Confirmed | |
|---|---|---|
| ☐ | ☒ | The exact sample size (*n*) for each experimental group/condition, given as a discrete number and unit of measurement |
| ☐ | ☒ | A statement on whether measurements were taken from distinct samples or whether the same sample was measured repeatedly |
| ☐ | ☒ | The statistical test(s) used AND whether they are one- or two-sided *Only common tests should be described solely by name; describe more complex techniques in the Methods section.* |
| ☐ | ☒ | A description of all covariates tested |
| ☐ | ☒ | A description of any assumptions or corrections, such as tests of normality and adjustment for multiple comparisons |
| ☐ | ☒ | A full description of the statistical parameters including central tendency (e.g. means) or other basic estimates (e.g. regression coefficient) AND variation (e.g. standard deviation) or associated estimates of uncertainty (e.g. confidence intervals) |
| ☐ | ☒ | For null hypothesis testing, the test statistic (e.g. *F*, *t*, *r*) with confidence intervals, effect sizes, degrees of freedom and *P* value noted *Give P values as exact values whenever suitable.* |
| ☐ | ☒ | For Bayesian analysis, information on the choice of priors and Markov chain Monte Carlo settings |
| ☒ | ☐ | For hierarchical and complex designs, identification of the appropriate level for tests and full reporting of outcomes |
| ☐ | ☒ | Estimates of effect sizes (e.g. Cohen's *d*, Pearson's *r*), indicating how they were calculated |

*Our web collection on statistics for biologists contains articles on many of the points above.*

## Software and code

Policy information about availability of computer code

| Data collection | N/A |
|---|---|
| Data analysis | R packages: BiomaRt v2.52; qqman v0.1.4; QCGWAS v1.0-8; gap v1.2.3-6; rGREAT v2.0.0; KEGGREST v1.36, coloc v3.1; HyPrColoc v1.0; oligo v1.62.0; limma v3.54.0; DESeq2 v1.38.0<br>Other: METAL v28.8.2018; LocusZoom v1.4; bedtools v2.27.0; GCTA v1.93.0beta; PhenoScanner v2; Variant Effect Predictor v98.3; STRINGdb v2.8.4 |

For manuscripts utilizing custom algorithms or software that are central to the research but not yet described in published literature, software must be made available to editors and reviewers. We strongly encourage code deposition in a community repository (e.g. GitHub). See the Nature Portfolio guidelines for submitting code & software for further information.

## Data

Policy information about availability of data

All manuscripts must include a data availability statement. This statement should provide the following information, where applicable:
- Accession codes, unique identifiers, or web links for publicly available datasets
- A description of any restrictions on data availability
- For clinical datasets or third party data, please ensure that the statement adheres to our policy

Full per-protein GWAS summary statistics are available for download at https://www.phpc.cam.ac.uk/ceu/proteins/ and the EBI GWAS Catalog https://

www.ebi.ac.uk/gwas/ (accession GCST90270765-GCST90270855). Individual-level genetic data cannot be shared in an open-access manner due to European and UK privacy (GDPR) legislation.

## Human research participants

Policy information about studies involving human research participants and Sex and Gender in Research.

| | |
|---|---|
| Reporting on sex and gender | We utilized data from population cohorts and case-control studies comprising both men and women, and did not perform any sex-specific analyses within this study. The findings from our study apply broadly to both men and women. Self-reported biological sex was used as a covariate in statistical models.  The terms "gender mismatch" or "sex mismatch" were used in Supplementary Table 1 to indicate where the biological sex of participants did not match self-reported sex - this was one of the exclusion criteria we applied. |
| Population characteristics | We utilized data from 10 studies, primarily comprising participants of European ancestry and both men and women.  All participants were adults (>=18).  Some studies were case-control designs, and so our study included patients with neurodegenerative and neuropsychiatric conditions, rheumatoid arthritis, coronary artery disease, stroke, or atrial fibrillation cases. Details of the participants are summarised in the Supplementary Tables and the Supplementary Note. |
| Recruitment | Our study utilized summary data from pre-existing studies only, so no new recruitment was performed.  As noted above, some studies comprised participants with chronic disease.  Where possible, diagnostic categories were included as covariates to minimise confounding. |
| Ethics oversight | Details of ethics oversight for each of the contributing studies are included in the Supplementary Note. |

Note that full information on the approval of the study protocol must also be provided in the manuscript.

# Field-specific reporting

Please select the one below that is the best fit for your research. If you are not sure, read the appropriate sections before making your selection.

☒ Life sciences          ☐ Behavioural & social sciences          ☐ Ecological, evolutionary & environmental sciences

For a reference copy of the document with all sections, see nature.com/documents/nr-reporting-summary-flat.pdf

# Life sciences study design

All studies must disclose on these points even when the disclosure is negative.

| | |
|---|---|
| Sample size | This discovery study was a meta-analysis of pre-existing pQTL GWAS summary statistics, and so the aim was to generate as large a sample as possible given available resources at the time the analyses were conducted.  As is typical for a GWAS meta-analysis, no power calculation was performed to inform sample size. |
| Data exclusions | Each contributing study had their own data exclusions, which included gender/sex mismatches, ethnic outliers, heterozygosity, cryptic relatedness, and low genotype call rates.  Duplicates were also detected and removed. Details are provided in the Supplementary Tables. |
| Replication | We performed replication and validation of our meta-analysis results using an independent cohort of 1,585 participants with pQTL data from the ARISTOTLE study. |
| Randomization | This is an observational study, rather than a clinical trial, so no randomisation was performed. |
| Blinding | This was not a clinical trial so there was no requirement for blinding. The nature of pQTL and other GWAS analyses in general render investigators blind to genotypic groups. |

# Reporting for specific materials, systems and methods

We require information from authors about some types of materials, experimental systems and methods used in many studies. Here, indicate whether each material, system or method listed is relevant to your study. If you are not sure if a list item applies to your research, read the appropriate section before selecting a response.

## Materials & experimental systems

| n/a | Involved in the study |
|---|---|
| ☒ | ☐ Antibodies |
| ☒ | ☐ Eukaryotic cell lines |
| ☒ | ☐ Palaeontology and archaeology |
| ☒ | ☐ Animals and other organisms |
| ☒ | ☐ Clinical data |
| ☒ | ☐ Dual use research of concern |

## Methods

| n/a | Involved in the study |
|---|---|
| ☒ | ☐ ChIP-seq |
| ☒ | ☐ Flow cytometry |
| ☒ | ☐ MRI-based neuroimaging |

