## [Peer Review File · Nature Immunology]

Peer Review Information

Journal: Nature Immunology

Manuscript Title: Genetics of circulating inflammatory proteins identifies drivers of immune-mediated disease risk and therapeutic targets

Corresponding author name(s): Dr. James Peters, Professor Adam Butterworth

Reviewer Comments & Decisions:

Decision Letter, initial version:
--

25th Apr 2023

Dear Dr Peters,

Your Article, "Mapping pQTLs of circulating inflammatory proteins identifies drivers of immune-related disease risk and novel therapeutic targets" has now been seen by 2 referees. You will see from their comments below that while they find your work of interest, some important points are raised. We are interested in the possibility of publishing your study in Nature Immunology, but would like to consider your response to these concerns in the form of a revised manuscript before we make a final decision on publication.

We therefore invite you to revise your manuscript taking into account all reviewer and editor comments. Please highlight all changes in the manuscript text file in Microsoft Word format.

* If you have not done so already please begin to revise your manuscript so that it conforms to our Article format instructions at <http://www.nature.com/ni/authors/index.html>. Refer also to any guidelines provided in this letter.

* Please include a revised version of any required reporting checklist. It will be available to referees to aid in their evaluation of the manuscript goes back for peer review. They are available here:

Reporting summary:

[REDACTED]

We hope to receive your revised manuscript within 4 weeks. If you cannot send it within this time, please let us know. We will be happy to consider your revision so long as nothing similar has been accepted for publication at Nature Immunology or published elsewhere.

Nature Immunology is committed to improving transparency in authorship. As part of our efforts in this direction, we are now requesting that all authors identified as 'corresponding author' on published papers create and link their Open Researcher and Contributor Identifier (ORCID) with their account on the Manuscript Tracking System (MTS), prior to acceptance. ORCID helps the scientific community achieve unambiguous attribution of all scholarly contributions. You can create and link your ORCID from the home page of the MTS by clicking on 'Modify my Springer Nature account'. For more information please visit www.springernature.com/orcid.

Sincerely,

Nick Bernard, PhD
Senior Editor
Nature Immunology

Reviewers' Comments:

Reviewer #1:

Remarks to the Author:

This is a multi-cohort GWAS with proteomics using the OLINK platform. OLINK provides multiple thematic panels, each covering around ninety proteins, and the Scallop consortium is organizing multi-cohort GWAS with these panels. This paper is the second one, covering 15,000 samples, after Folkersen et al., Nat Metab. 2020 with 30,000 samples. The GWAS is excellently executed and state-of-the-art, and I really have nothing to criticize there. It's a great example of how these analyses should be done. The paper takes up interesting associations for discussion and puts them into their biological context. As a whole this paper constitutes a great resource for the community and can be published as-is.

Reviewer #2:

Remarks to the Author:

Start of the results, are the 180 pQTL associations independent? i.e. have they been clumped or similar?

I found the replication analysis fairly underwhelming, do the authors have an explanation as to why only 72 pQTLs have $P < 2.9 \times 10^{-4}$? If they believe this may be due to the relatively modest sample size of the ARISTOTLE trial, would it not be worthwhile evaluating these pQTL in a larger sample e.g. ref 10 from their study?

Do the authors have an explanation for the surprising lack of genetic colocalization between their cis-pQTL and cis-eQTL derived from the eQTLGen project?

Is using $r^2 > 0.8$ being sentinel pQTL and GWAS hit really sufficient to claim that MR findings are 'unlikely to be due to confounding by LD'? Why was genetic colocalization not applied here as it was to evaluate shared variants with gene expression in this work?

Also was LD accounted for when selecting IVs for MR given the $r^2 < 0.1$ threshold?

Author Rebuttal to Initial comments

See Inserted PDF

Response to reviewers

We would like to thank the reviewers for their reviews and constructive comments and suggestions. We have addressed the specific points, including adding new analyses where appropriate, as detailed below. Reviewer comments are in *blue italics*.

Where we quote passages of the text, we indicate this with pale pink highlighting.

We have highlighted changes in the manuscript in blue to allow easy identification of changes.

For ease of reference, the Extended Data Figures are contained within the Supplementary Information pdf.

Reviewers' Comments:

Reviewer #1:

Remarks to the Author:

This is a multi-cohort GWAS with proteomics using the OLINK platform. OLINK provides multiple thematic panels, each covering around ninety proteins, and the Scallop consortium is organizing multi-cohort GWAS with these panels. This paper is the second one, covering 15,000 samples, after Folkersen et al., Nat Metab. 2020 with 30,000 samples. The GWAS is excellently executed and state-of-the-art, and I really have nothing to criticize there. It's a great example of how these analyses should be done. The paper takes up interesting associations for discussion and puts them into their biological context. As a whole this paper constitutes a great resource for the community and can be published as-is.

We thank the reviewer for their positive feedback and enthusiastic support for the manuscript.

Reviewer #2:

Remarks to the Author:

j) Start of the results, are the 180 pQTL associations independent? i.e. have they been clumped or similar?

Author response

In brief, yes, the associations are independent due to the region-based locus definition we initially use. 180 represents the number of locus-protein associations. For a given protein, these associations are therefore independent due to distance. We defined our method for locus definition in the Methods (see below in quotes).

“Definition of pQTL sentinel variants and regions. We defined a pQTL as a genetic locus significantly ($P < 5 \times 10^{-10}$) associated with protein abundance. We defined the sentinel variant at a locus as the variant with the lowest P-value in the region for a given protein. We used the

following approach for each protein to define genomic regions and the sentinel variant in each: 1) we first obtained a list of significant ($P < 5 \times 10^{-10}$) variants and the flanking region (± 1 Mb) for each variant; 2) overlapping regions were then iteratively merged until no overlapping regions remained; 3) the most significant variant in each resulting region was then defined as the sentinel variant. This approach has the flexibility to cope with long stretches of LD whilst avoiding the drawback of setting a longer than necessary region for all variants. The algorithm was implemented using bedtools v2.27.0. Signals within/beyond 1Mb of the transcription start site (TSS) of the gene encoding the target protein were defined as *cis* and *trans*, respectively.”

To improve clarity we have edited the Results as follows (changes in blue font):

“We identified a total of 180 significant ($P < 5 \times 10^{-10}$) associations between 108 genomic regions (see **Methods** for locus definition) and 70 proteins (**Figure 1, Supplementary Table 3, Supplementary Item, Supplementary Figures 1-2**). To date, 50 of these associations have not previously been reported in peer-reviewed articles ($r^2 \geq 0.8$)^{1, 5-12}(**Table 1**). Of the 180 significant locus-protein associations, 59 (33%) were local-acting (*cis* pQTLs; defined here as a genetic variant lying within ± 1 megabase of the gene encoding the associated protein) and 121 (67%) were distant-acting (*trans*).”

Within a locus, there may be more than one independent pQTL signal, which were not counted in the total of 180 locus-protein associations. Indeed, using conditional analysis with GCTA-COJO we identified an additional 47 secondary signals, as described at the end of the second paragraph of the Results and pasted below.

“Conditional analyses using GCTA-COJO (**Methods**) revealed the presence of an additional 47 independent signals, which were mostly *cis*. This raised the total number of pQTL signals from 180 (59 *cis*, 108 *trans*) to 227 (99 *cis*, 128 *trans*) (**Supplementary Table 4**).”

ii) I found the replication analysis fairly underwhelming, do the authors have an explanation as to why only 72 pQTLs have $P < 2.9 \times 10^{-4}$? If they believe this may be due to the relatively modest sample size of the ARISTOTLE trial, would it not be worthwhile evaluating these pQTL in a larger sample e.g. ref 10 from their study?

Author Response

As the reviewer alludes to, our primary dataset was nearly ten times larger than the replication cohort sample size ($n=15,150$ versus $1,585$, respectively). It is perhaps therefore unsurprising that around only 40% of pQTLs replicated at a Bonferroni-corrected significance level due to lack of power in the replication cohort. However, we highlight the very high concordance in terms of effect sizes (Pearson $r=0.97$, **Extended Data Figure 3** of the revised manuscript, also shown below) indicating that the effects seen in our replication cohort are highly consistent with our primary analysis.

In addition, we highlight that we already implemented a filtering step in our primary meta-analysis to ensure consistency across individual cohorts. Thus, our primary meta results have a ‘pseudo-replication step’ built in. The relevant section from the Methods is pasted below.

“To remove potentially erroneous meta-analysis signals arising due to an extremely strong association in a single cohort, we examined the meta-analysis results at each sentinel variant by visual inspection of the forest plot and imposed the following criteria: 1) to be included in the meta-analysis, a variant was required to be available in at least three studies and in at least 3,500 participants; 2) in order to be declared significant, we required a meta-analysis $P < 5 \times 10^{-10}$, and, if there was evidence of heterogeneity with $I_2 > 30\%$, then we required the P-value in at least three studies to be < 0.05 and the direction of effect in those studies to be consistent with the overall meta-analysis results.”

Extended Data Figure 3. pQTL replication in the ARISTOTLE cohort. Comparison of effect sizes between pQTLs from the discovery pQTL meta-analysis (n=15,150) and the ARISTOTLE cohort (n=1,585). Each point represents a genetic variant that was a significant pQTL in the discovery meta-analysis. Effect size = standard deviation (sd) increase in protein per allele. 174 of 180 genetic variants were available for testing in the ARISTOTLE data. Red= cis, Blue= trans.

At the reviewer’s suggestion, we also compared our results to those from the study by the deCODE group (Ferkingstad et al, Nature Genetics 2021 DOI: [10.1038/s41588-021-00978-w](https://doi.org/10.1038/s41588-021-00978-w)), hereafter referred to as the “deCODE study”. This study performed pQTL mapping in approximately 35,000 individuals from Iceland, with proteins measured using the aptamer-

based SomaScan V4 platform (whereas we used the antibody-based Olink Inflammation platform in our study).

72 of the 91 proteins in our study were measured in the deCODE study. Of the 180 significant locus-protein associations from our study, 158 were testable for replication in the deCODE summary statistics. Of these 158 associations, 78 were significant at $P < 5 \times 10^{-8}$. Using a more liberal p-value threshold of 2.8×10^{-4} (a Bonferroni-correction for the 180 pQTLs), 96 were significant. Overall, we replicated 126 (71%) of the 178 testable pQTLs in either ARISTOTLE or deCODE at $P < 2.8 \times 10^{-4}$ (see new **Supplementary Note Table 1**).

We noted that 7 pQTLs replicated in ARISTOTLE with $P < 5 \times 10^{-8}$ and yet did not replicate in the deCODE study despite the substantially larger sample size of the latter study (see new **Supplementary Note Table 1**). These included 4 cis pQTLs, as well as two trans pQTLs described in previous studies that used single protein ELISAs, including a well-known trans pQTL in the *IL6R* gene affecting IL-6 levels. The identification of these two trans pQTLs in our Olink data and external studies that used ELISAs, but not with the Somascan assay, suggests the lack of replication in deCODE might relate to a platform effect.

We have added details of this new analysis to the main text (third paragraph of the Results, lines 90-96) and the Supplementary Information.

iii) Do the authors have an explanation for the surprising lack of genetic colocalization between their cis-pQTL and cis-eQTL derived from the eQTLGen project?

Author response:

First we want to point out and apologise for an error in the first version of our manuscript. We originally stated that 32 of the 59 cis pQTLs also had an genome-wide significant eQTL in eQTLGen. In fact, the correct number is 40. One additional gene (*TGFB1*) had a cis eQTL that did not reach genome-wide significance, but that was significant at 5% FDR in the eQTLGen data.

We thank the reviewer for pointing out this interesting observation that of these 40 genome-wide significant blood cis eQTLs in eQTLGen, we only observed robust evidence of colocalisation (PP H4>0.8) with the corresponding cis pQTL in 6 instances (**Supplementary Table 6**). An explanation for this may be that the plasma proteome is not the direct corollary of the whole blood transcriptome. First, there is a difference in physical compartment: the plasma proteome predominantly reflects soluble extracellular proteins whereas the blood transcriptome is intracellular. Thus variants affecting protein cleavage from the cell surface or secretion can impact circulating protein levels. Second, a wide range of tissues other than blood are the primary source of many plasma proteins. This is evident when considering circulating proteins that are measured as biomarkers in clinical practice (e.g. albumin produced by the liver, troponin by the heart, PSA by the prostate). A further possibility is that some pQTLs are the result of alternative splicing rather than an eQTL (e.g. an alternative isoform is preferentially cleaved from the cell surface into the plasma).

To further explore the potential for colocalisation of eQTLGen (whole blood) cis eQTLs and circulating pQTLs, we have provided regional association ('LocusZoom') plots comparing the pQTL and eQTL signals (new **Supplementary Figure 3**). Visual inspection of these plots suggests that while the results of colocalization testing may be slightly conservative in some instances, the majority of cis pQTL signals do not line up with the blood eQTLs. Some clear visual examples of this include ADA, CSF1, CXCL6, CXCL9, S100A12, HGF, IL7, LAP, CCL8.

When we extend our eQTL-pQTL colocalization across multiple tissues in GTEx and the eQTL Catalog we find strong evidence of colocalization (PP H4 >0.8) for 32 of the 59 cis pQTLs with the cognate eQTL in at least one cell or tissue type. At the more liberal threshold of PP H4 >0.5, 38 cis eQTLs colocalise. These data support the hypothesis that some eQTLs underpinning plasma pQTLs are in tissues other than blood (**Supplementary Tables 7-8**).

We have amended the Results (lines 124-125, 131-136) and the Discussion (lines 366-383) to reflect the points listed above.

iv) Is using $r^2 > 0.8$ being sentinel pQTL and GWAS hit really sufficient to claim that MR findings are 'unlikely to be due to confounding by LD'? Why was genetic colocalization not applied here as it was to evaluate shared variants with gene expression in this work?

Author response:

In our experience, $r^2 > 0.8$ between the lead variants for two traits is a strong proxy for colocalisation. However, we accept the reviewer's point that further evaluation, including a formal test of colocalisation is preferable. Therefore, for the 12 significant protein-disease pairs that resulted from our previous MR filtering steps, we have i) provided regional association ("LocusZoom") plots to allow visual comparison of the cis pQTL and relevant disease signals (**Supplementary Figure 4** of the revised manuscript), and ii) added an additional step of performing colocalisation between the cis pQTL association signal and the disease signal using the PWCoCo ('Pair-Wise Conditional analysis and Colocalisation analysis') package (Robinson *et al.* 2022. An efficient and robust tool for colocalisation: Pair-wise Conditional and Colocalisation (PWCoCo). bioRxiv: doi: <https://doi.org/10.1101/2022.08.08.503158>). This revealed strong evidence (PP H4 >0.8) for colocalisation for 10 of the pairs, and moderate evidence in a further instance (PP H4 >0.5 but <0.8), suggesting that $r^2 > 0.8$ is indeed acting as a reliable proxy for formal colocalisation (see table below). The cis pQTL for IL12B (a subunit of IL12) and Crohn's disease GWAS signal did not colocalise (PP H4 0.06) and so we have amended the manuscript to reflect this. Of note, the cis pQTL for IL12B *did* colocalise with the overall inflammatory bowel disease (IBD) GWAS signal (i.e. Crohn's disease and ulcerative colitis analysed together versus controls), as well as the ulcerative colitis (UC) signal. Crohn's disease is the most common form of IBD, and the IBD GWAS thus contains more Crohn's disease cases than UC cases. In addition, ustekinumab (anti-IL12/23 therapy) is effective in both CD and UC, supporting the biological plausibility of the MR result for Crohn's disease based on the $r^2 > 0.8$ filter.

Protein	Disease	PP H4	Comment
CD40	Crohn's disease	0.92	PWCoCo
CD40	Inflammatory bowel disease	0.97	
CD40	Multiple sclerosis	0.97	
CD40	Rheumatoid arthritis	0.99	
CD5	Primary sclerosing cholangitis	0.88	
CD6	Inflammatory bowel disease	0.99	
CXCL5	Ulcerative colitis	0.98	
IL-12B	Crohn's disease	0.06	PWCoCo
IL-12B	Inflammatory bowel disease	0.82	PWCoCo
IL-12B	Ulcerative colitis	0.99	PWCoCo
IL-18R1	Crohn's disease	0.57	
IL-18R1	Eczema	0.94	

Changes:

-We have added the results of the colocalisation result in the table above to **Supplementary Table 14**.

-We have amended the Results (lines 295-301) and Methods (lines 864-877) to reflect this analysis and its findings.

v) Also was LD accounted for when selecting IVs for MR given the $r^2 < 0.1$ threshold?

Author response:

We thank the reviewer for the opportunity to clarify this point. As the reviewer points out, the $r^2 < 0.1$ threshold that we used allows inclusion of weakly correlated genetic variants in the MR instruments. We selected this threshold because we used the GSMR method which accounts for LD among instruments when these are not fully independent, as detailed in the paper by

(Zhu et al, Nature Communications 2018 <https://doi.org/10.1038/s41467-017-02317-2>). In brief, LD between a pair of genetic variants i and j is included in the calculation of the variance-covariance matrix of $b(\hat{)}_{xy}$ (the estimate of the causal effect of exposure x on outcome y).

Zhu et al's study provides evidence from simulations that in the presence of LD (pruning using $r^2 < 0.5$), i) the test statistics are well-calibrated under the null hypothesis of no association between the exposure and the outcome, and ii) the estimate of b_{xy} is unbiased under the alternative hypothesis that there is a causal effect of x on y (please see the Supplementary material, Supplementary Figure 1b and Supplementary Table 1 of Zhu et al's publication for full details).

We have amended the Methods to make it clear that GSMR accounts for LD between instruments (lines 843-844).

Additional non-reviewer driven changes:

1) We identified a discrepancy in the reporting of the number of pQTLs colocalising with eQTLs from GTEx and the eQTL Catalogue. This discrepancy resulted from two issues. First, H4/H3 ratio had been used to report colocalisation in the eQTL Catalogue analysis instead of H4 > 0.8 as described in the text and used in the rest of the colocalisation analyses. We have fixed this so that our approach is now consistent across all analyses.

Second, no coloc results were displayed for a small number of gene/protein pairs due to use of deprecated Ensembl IDs in the comparison of pQTLs to GTEx and eQTL Catalogue data.

Revisions to the text:

“Systematic COLOC analyses revealed colocalising (PP > 0.8) *cis*-eQTLs in at least one tissue or cell type for **32** <previously reported as 30> of the 59 *cis*-pQTLs (**Supplementary Tables 7-8**); **16** <previously 15> were highlighted by both eQTL resources, **12** <previously 10> by GTEx only, and the remaining 4 <previously 5> by the eQTL Catalogue only.”

2) Reformatting to meet Nature Immunology requirements (10 Extended Data Figures, plus additional Supplementary Information, and re-phrasing more concisely to reduce word count).

3) Minor revision to the title changing “immune-related diseases” to “immune-mediated diseases”.

Decision Letter, first revision:

Our ref: NI-A35592A

7th Jun 2023

Dear Dr. Peters,

Thank you for submitting your revised manuscript "Mapping pQTLs of circulating inflammatory proteins identifies drivers of immune-mediated disease risk and novel therapeutic targets" (NI-A35592A).

It has now been seen by the original referee number 2. Please note that we did not need to go back to reviewer 1, but we did recruit a new reviewer (#3) as we thought that we made a mistake initially in that reviewers 1 and 2 were very strong genetics reviewers but not really immunologists, and so reviewer 3 is an immunologist. As you can see, reviewer 3 has some textual revisions requested but generally is happy with the paper.

Overall the reviewers find that the paper has improved in revision, and therefore we'll be happy in principle to publish it in Nature Immunology, pending minor revisions to satisfy the referees' final requests and to comply with our editorial and formatting guidelines.

We will now perform detailed checks on your paper and will send you a checklist detailing our editorial and formatting requirements in about a week. Please do not upload the final materials and make any revisions until you receive this additional information from us.

If you had not uploaded a Word file for the current version of the manuscript, we will need one before beginning the editing process; please email that to immunology@us.nature.com at your earliest convenience.

Thank you again for your interest in Nature Immunology Please do not hesitate to contact me if you have any questions.

Sincerely,

Nick Bernard, PhD
Senior Editor
Nature Immunology

Reviewer #2 (Remarks to the Author):

Thank you for thoroughly addressing my comments. I have no further suggestions.

Reviewer #3 (Remarks to the Author):

It was a pleasure to have the opportunity to review this manuscript. Overall I found it well written and accessible. I was asked to review the sections relevant to immunology and based on this my comments are focused on this area with a few specific comments on the discussion of the findings.

- 1) Overall I believe it is important for the authors to highlight the fact that plasma protein levels may be reflective of an immune process but the directionality of expression may be different in the plasma than at the site of inflammation. The authors note this early in the manuscript and it may be repeating in the discussion.
- 2) Lines 437-441 - make an important point about commonalities in pathogenesis between diseases being revealed through this approach- this is useful but I would omit specific links that are given in the parenthesis as too detailed and unsupported.
- 3) The section on CXCL5 and UC is lengthy and a bit overextended. In line 467-469 the authors suggest that "differences in the processes of disease initiation and perpetuation" can be discerned from the MR analysis - I do not believe they have shown this.
- 4) Comments in line 502-504 related to RA and development of therapies is overreaching with respect to its usefulness and I would suggest modifying it
- 5) the discussion of CD40 levels and host defences in the gut (line 518-520) is overreaching for significance and I would leave out or modify

Author Rebuttal, first revision:

Author response to reviewers

We thank the reviewers for their comments. We have revised the manuscript to reflect the suggested edits.

Reviewer #3:

Remarks to the Author:

It was a pleasure to have the opportunity to review this manuscript. Overall I found it well written and accessible. I was asked to review the sections relevant to immunology and based on this my comments are focused on this area with a few specific comments on the discussion of the findings.

- 1) Overall I believe it is important for the authors to highlight the fact that plasma protein levels may be reflective of an immune process but the directionality of expression may be different in the plasma than at the site of inflammation. The authors note this early in the manuscript and it may be repeating in the discussion.

We agree with the reviewer's point. We have added to the Discussion (new text in purple):

"Finally, as with all epidemiological-scale pQTL studies, proteins were measured in plasma (i.e. the extracellular component of blood), which may not always be the disease-relevant biological compartment, and where the direction of genotype-expression association may even be opposite to the site of inflammation. Thus, future tissue- and cell-specific pQTL studies will be valuable to understand differences in genetic signals across tissues."

2) Lines 437-441 - make an important point about commonalities in pathogenesis between diseases being revealed through this approach- this is useful but I would omit specific links that are given in the parenthesis as too detailed and unsupported.

We have now removed the relevant comments.

3)The section on CXCL5 and UC is lengthy and a bit overextended. In line 467-469 the authors suggest that "differences in the processes of disease initiation and perpetuation" can be discerned from the MR analysis - I do not believe they have shown this.

We thank the reviewer for highlighting this. We have shortened and restructured the sections on CXCL5 and UC to improve clarity in both the Results and the Discussion.

To clarify, we did not claim that differences in the processes of disease initiation and perpetuation could be discerned from the MR analysis, but rather raised this possibility as a hypothesis in the Discussion that might account for the discordance between the MR analysis and the UC vs healthy control differential expression analysis. The previous version of our manuscript stated:

"We hypothesize that the discrepancy between direction of effects from the MR analysis and from analysis of patient samples might reflect differences in the processes of disease initiation and perpetuation."

Nevertheless, we appreciate that this section could have been better written and more clearly delineate our findings versus those of previous studies. In summary, our MR analysis shows that genetic tendency to higher levels of CXCL5 in plasma is protective against UC (reduced risk of disease), and this observation also holds true at the mRNA level in both whole-blood and colon tissue. Alternatively, this can be expressed as genetic tendency to lower levels of CXCL5 as increasing risk of UC. In contrast, our differential expression analysis in gut tissue shows CXCL5 is higher in UC than in controls. The reviewer is correct to say that we cannot infer from our data that CXCL5 is driving disease perpetuation or severity. However, evidence from two other recent studies provide at least circumstantial evidence for this. Friedrich *et al* (Nature Medicine 2021) identified a gene module containing neutrophil chemokine genes including CXCL5 that was associated with disease severity and reduced responsiveness to treatment. Clearly, this is observational data and a causal relationship to severity or outcome is not proven by their data. Pavlidis *et al* (Nature Communications 2022) made similar observations using UC patient samples but also showed that targeting CXCR2 (the receptor for CXCL5) ameliorates disease severity in multiple animal models of UC. We have revised the Discussion to make this section clearer (relevant section pasted below in purple).

It is interesting that this apparent contradiction between the MR analysis and other lines of evidence has parallels with another example in IBD, namely TL1A (*TNFSF15*). Genotypes conferring lower *TNFSF15* expression are a risk factor for disease, but TLA1 levels are increased

both locally and systemically in IBD patients, and anti-TL1A therapies have shown efficacy in recent phase 2 trials.

“Our MR findings implicate CXCL5 in the aetiology of UC, where genetic susceptibility to higher levels of plasma CXCL5 was associated with lower UC risk. Examination of eQTL data revealed this observation was consistent at the RNA level in both the blood and gut tissue. By contrast, in our case-control analysis comparing gut tissue from UC patients versus controls, *CXCL5* is one of the most upregulated transcripts. A previous study reported that serum levels of CXCL5 are higher in IBD patients than controls⁴². Recent studies using UC gut tissue have implicated upregulation of genes encoding neutrophil-targeting chemokines, including *CXCL5*, by non-immune cells as correlating with important histopathological features, such as ulceration, and differentiating patient trajectories, including their responsiveness to different treatments^{43,44}. Targeting CXCR2, the receptor for CXCL5, significantly attenuates animal models of UC⁴⁴. One possible explanation that may reconcile these apparently contradictory findings is that genetic tendency to lower CXCL5 expression increases UC risk through impaired mucosal immune homeostasis, but that elevated CXCL5 is an important driver of tissue injury once disease is initiated. By analogy, a non-coding genetic variant associated with lower gene and protein expression of *TNFSF15* (encoding the inflammatory cytokine TL1A) in monocytes and macrophages increases IBD susceptibility⁴⁵, but TL1A is upregulated both systemically and in the gut in patients with active IBD{Bamias, 2012 #1805;Bamias, 2010 #1806;} and anti-TL1A therapies have recently shown efficacy in IBD in phase 2 randomised trials (NCT05013905 and NCT04996797⁴⁶).”

4) Comments in line 502-504 related to RA and development of therapies is overreaching with respect to its usefulness and I would suggest modifying it.

We have removed this given the need to fit the word count and the fact these comments are speculative rather than firmly supported by our data.

5) the discussion of CD40 levels and host defences in the gut (line 518-520 is overreaching for significance and I would leave out or modify.

We have removed this.

Other changes:

In addition, we have edited the manuscript to meet journal specifications, including shortening the main text and Methods to 5,000 and 3,000 words, respectively, and removing references to “novel/new/first” when describing our findings. Important supporting text that did not fit within the word limit has been moved to Supplementary material.

Previous Table 1 did not fit the criteria for a main display item, and we have replaced this with a new Table 1 summarising the Mendelian randomisation analysis. The information in the previous Table 1 can be found in Supplementary Tables 3 and 9.

We noted a minor error in the previous manuscript in the sample sizes for two cohorts (due to reporting sample size with proteomic data rather than the sample size with both proteomic and genetic data). The sample sizes have been corrected to report the numbers of samples used in the pQTL analysis. The total sample size is 14,824 participants.

Final Decision Letter:

Dear Dr. Peters,

I am delighted to accept your manuscript entitled "Genetics of circulating inflammatory proteins identifies drivers of immune-mediated disease risk and therapeutic targets" for publication in an upcoming issue of Nature Immunology.

Over the next few weeks, your paper will be copyedited to ensure that it conforms to Nature Immunology style. Once your paper is typeset, you will receive an email with a link to choose the appropriate publishing options for your paper and our Author Services team will be in touch regarding any additional information that may be required.

Please note that *Nature Immunology* is a Transformative Journal (TJ). Authors may publish their research with us through the traditional subscription access route or make their paper immediately open access through payment of an article-processing charge (APC). Authors will not be required to make a final decision about access to their article until it has been accepted. [Find out more about Transformative Journals](https://www.springernature.com/gp/open-research/transformative-journals).

Authors may need to take specific actions to achieve [compliance with funder and institutional open access mandates](https://www.springernature.com/gp/open-research/funding/policy-compliance-faqs). If your research is supported by a funder that requires immediate open access (e.g. according to [Plan S principles](https://www.springernature.com/gp/open-research/plan-s-compliance)) then you should select the gold OA route, and we will direct you to the compliant route where possible. For authors selecting the subscription publication route, the journal's standard licensing terms will need to be accepted, including [self-archiving policies](https://www.springernature.com/gp/open-research/policies/journal-policies). Those licensing terms will supersede any other terms that the author or any third party may assert apply to any version of the manuscript.

Your paper will be published online soon after we receive your corrections and will appear in print in the next available issue. Content is published online weekly on Mondays and Thursdays, and the embargo is set at 16:00 London time (GMT)/11:00 am US Eastern time (EST) on the day of publication. Now is the time to inform your Public Relations or Press Office about your paper, as they might be interested in promoting its publication. This will allow them time to prepare an accurate and satisfactory press release. Include your manuscript tracking number (NI-A35592B) and the name of the journal, which they will need when they contact our office.

About one week before your paper is published online, we shall be distributing a press release to news organizations worldwide, which may very well include details of your work. We are happy for your institution or funding agency to prepare its own press release, but it must mention the embargo date and *Nature Immunology*. Our Press Office will contact you closer to the time of publication, but if you or your Press Office have any enquiries in the meantime, please contact press@nature.com.

Also, if you have any spectacular or outstanding figures or graphics associated with your manuscript - though not necessarily included with your submission - we'd be delighted to consider them as candidates for our cover. Simply send an electronic version (accompanied by a hard copy) to us with a possible cover caption enclosed.

If you have not already done so, we strongly recommend that you upload the step-by-step protocols used in this manuscript to the Protocol Exchange. Protocol Exchange is an open online resource that allows researchers to share their detailed experimental know-how. All uploaded protocols are made freely available, assigned DOIs for ease of citation and fully searchable through nature.com. Protocols can be linked to any publications in which they are used and will be linked to from your article. You can also establish a dedicated page to collect all your lab Protocols. By uploading your Protocols to Protocol Exchange, you are enabling researchers to more readily reproduce or adapt the methodology you use, as well as increasing the visibility of your protocols and papers. Upload your Protocols at www.nature.com/protocolexchange/. Further information can be found at www.nature.com/protocolexchange/about .

Please note that we encourage the authors to self-archive their manuscript (the accepted version before copy editing) in their institutional repository, and in their funders' archives, six months after publication. Nature Portfolio recognizes the efforts of funding bodies to increase access of the research they fund, and strongly encourages authors to participate in such efforts. For information about our editorial policy, including license agreement and author copyright, please visit www.nature.com/ni/about/ed_policies/index.html

Sincerely,

Nick Bernard, PhD
Senior Editor
Nature Immunology